# Structural basis of the human NAIP/NLRC4 inflammasome assembly and pathogen sensing

Rosalie E. Matico[1,6], Xiaodi Yu ✉[1,6], Robyn Miller[1], Sandeep Somani [2], M. Daniel Ricketts[1], Nikit Kumar[1], Ruth A. Steele [1], Quintus Medley[3], Scott Berger[4], Benjamin Faustin[5] & Sujata Sharma[1]

The NLR family caspase activation and recruitment domain-containing 4 (NLRC4) inflammasome is a critical cytosolic innate immune machine formed upon the direct sensing of bacterial infection and in response to cell stress during sterile chronic inflammation. Despite its major role in instigating the subsequent host immune response, a more complete understanding of the molecular events in the formation of the NLRC4 inflammasome in humans is lacking. Here we identify *Bacillus thailandensis* type III secretion system needle protein (Needle) as a potent trigger of the human NLR family apoptosis inhibitory protein (NAIP)/NLRC4 inflammasome complex formation and determine its structural features by cryogenic electron microscopy. We also provide a detailed understanding of how type III secretion system pathogen components are sensed by human NAIP to form a cascade of NLRC4 protomer through a critical lasso-like motif, a 'lock–key' activation model and large structural rearrangement, ultimately forming the full human NLRC4 inflammasome. These results shed light on key regulatory mechanisms specific to the NLRC4 inflammasome assembly, and the innate immune modalities of pathogen sensing in humans.

The nucleotide-binding domain and leucine-rich repeat-containing (NLR) proteins are cytosolic proteins involved in host innate immune responses to infections and cell stress[1]. After sensing cellular cues, NLRs (including the NLR family caspase activation and recruitment domain (CARD) domain-containing 4, NLRC4) assemble into a molecular scaffold that recruits and processes inflammatory caspases (including caspase-1) by a close-proximity mechanism, forming the so-called inflammasome cßomplexes. Active caspase-1 can then mediate numerous downstream immune signaling events including the maturation of inflammatory cytokines, pyroptotic cell death, production of inflammatory lipids and modulation of cellular metabolism. Especially in humans, the NLR family apoptosis inhibitory protein (NAIP)/NLRC4 inflammasome has been shown to be triggered after the sensing of bacterial type III secretion systems (T3SS) during infection[2,3], as well as in sterile conditions during human aging through metabolic dysregulation[4] and endogenous retrotransposons in auto-immune diseases[5]. Importantly, there are constitutive NLRC4 hyperactivation through Mendelian inheritance of various de novo gain-of-function mutations that form an expanding list of autoinflammatory syndromes collectively termed NLRC4 inflammasomopathies[6]. Despite this strong relevance in host immune response to infection and inflammatory diseases, the detailed structural basis of the NLRC4 inflammasome assembly and pathogen sensing in humans was lacking.

[1]Structural and Protein Sciences, Johnson & Johnson Innovative Medicine, Spring House, PA, USA. [2]In Silico Discovery Sciences, Johnson & Johnson Innovative Medicine, Spring House, PA, USA. [3]Discovery Immunology, Johnson & Johnson Innovative Medicine, Cambridge, MA, USA. [4]Discovery Immunology, Johnson & Johnson Innovative Medicine, Spring House, PA, USA. [5]Discovery Immunology, Johnson & Johnson Innovative Medicine, San Diego, CA, USA. [6]These authors contributed equally: Rosalie E. Matico, Xiaodi Yu. ✉e-mail: xyu6@its.jnj.com

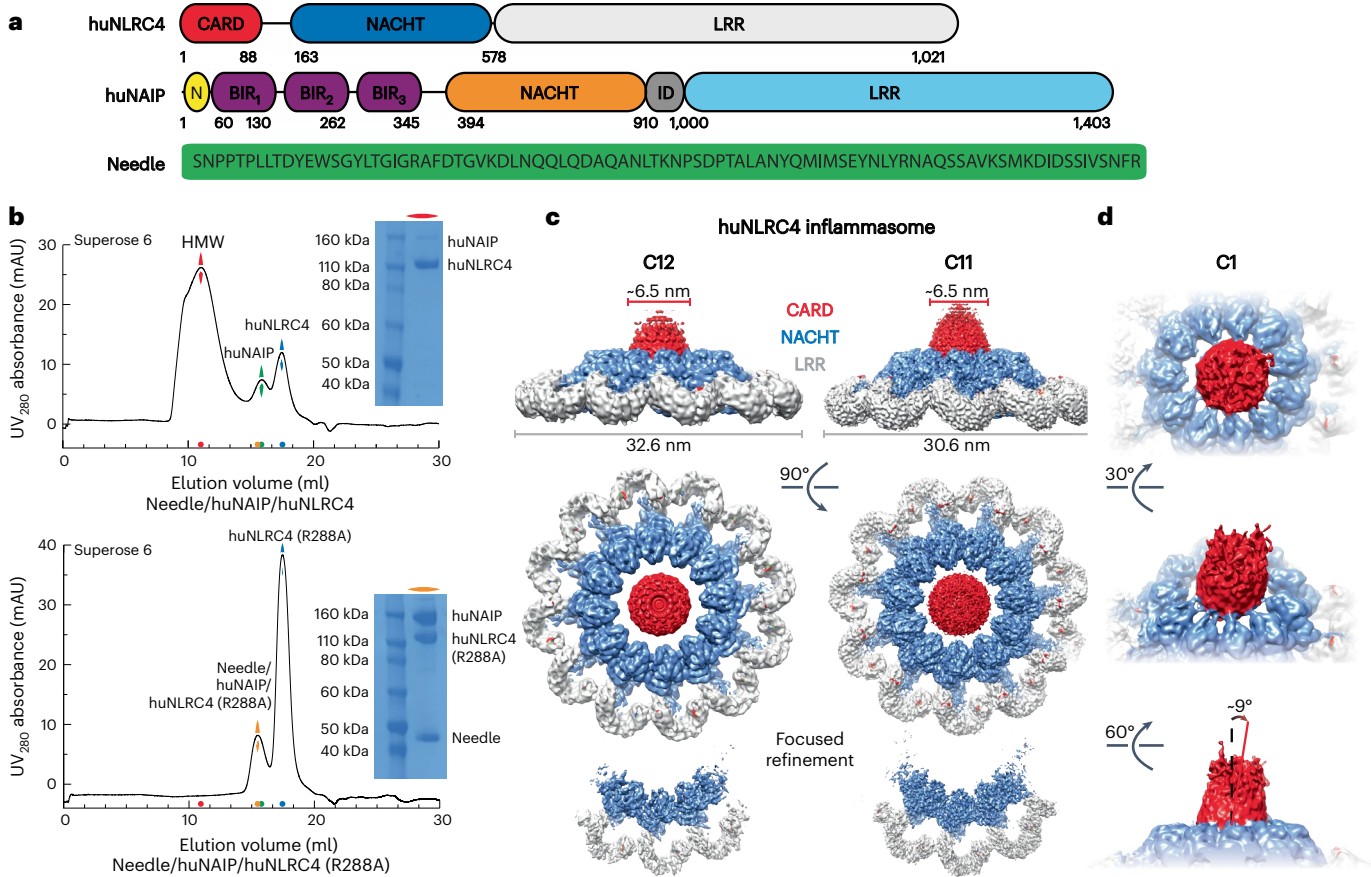

**Fig. 1 | Biochemical reconstitution and cryo-EM structure of the full human NAIP/NLRC4 inflammasome. a**, Domain organization of huNLRC4, huNAIP and Needle. **b**, Purification of huNLRC4 inflammasome with or without R288A mutation and SDS–PAGE of SEC fractions from the peaks (highlighted with bars). These experiments were independently repeated three times. **c**, Cryo-EM maps of huNLRC4 inflammasome. **d**, Zoom-in views of CARD domains. Maps are colored by domain organization.

## Results

### Human NAIP/NLRC4 inflammasome reconstitution

Previous studies using murine systems have shown that mouse NLRC4 (mNLRC4) activation requires various sensor proteins (NAIP1–6) that recognize different bacterial T3SS components known as triggers (for example, flagellin, Needle or rod)[7–12], whereas only one NAIP is present in humans (huNAIP)[13]. The binding of the trigger activates mouse NAIP (mNAIP), exposing a catalytic surface that binds a receptor surface of mNLRC4 resulting in a cascade event that forms the inflammasome. The amino acid (a.a.) residues of the catalytic surface in the mouse model identified key residues needed for this activation step[7,11,14,15]. The mutation of residue Arg288 to alanine on mNLRC4 resulted in stunted inflammasome formation indicating that it is a critical residue on the catalytic surface[16]. Two inflammasome structures using truncated mNLRC4 have been elucidated following purification from activated cells, however, a full-length (FL) human NLRC4 (huNLRC4) inflammasome structure has yet to be reported. In this Article, we describe the modalities of complex assembly subsequent to pathogen ligand sensing, and the resulting high-resolution single-particle cryogenic electron microscopy (cryo-EM) structures of ligand sensing by huNAIP and the full human NAIP/NLRC4 inflammasome complex.

To elucidate the biochemical requirements for inflammasome assembly, we started by expressing and purifying the various human FL components using the baculoviral expression system including recombinant His–Lfn–Needle (NeedleTox), huNAIP and huNLRC4 with or without R288A mutation (Fig. 1a,b and Extended Data Fig. 1). Initially, each separated component eluted as monomers by size exclusion chromatography (SEC), but when mixed together in a 1:1:10 (NeedleTox:huNAIP:huNLRC4) molecular ratio we observed a single high-molecular-weight (HMW) peak that eluted close to the void volume (apparent molecular weight >1 MDa, Fig. 1b). We confirmed by sodium dodecyl sulfate–polyacrylamide gel electrophoresis (SDS–PAGE) that this peak contained all three components, with huNLRC4 being the dominant band and present at a notable higher ratio to NAIP and Needle. In contrast, use of huNLRC4-R288A mutant in the mixture induced a complex that eluted later than the wild-type (WT) Needle/huNAIP/huNLRC4 complex in SEC with an apparent molecular weight ~350 kDa. The SDS–PAGE of the SEC peak revealed that this complex was composed of NeedleTox, huNAIP and huNLRC4-R288A in equal molar ratios. These results indicated that purified NeedleTox, huNAIP and huNLRC4 were competent to assemble into a full inflammasome complex. Comparable to mNLRC4-R288A where cellular co-expression of the murine components inhibited full inflammasome assembly, huNLRC4-R288A also impeded full assembly generating a stable Needle/huNAIP/huNLRC4-R288A ternary complex.

### Structures of Needle/huNAIP/huNLRC4 inflammasome complex

The HMW peak fraction from Needle/huNAIP/huNLRC4 mixture was applied directly to the cryo-EM grids. In the raw electron microscopy (EM) images, aggregated particles were predominant, but disk-like

**Table 1 | Cryo-EM data collection, refinement and validation statistics**

| | huNLRC4 C11 complex (EMDB-29496, PDB 8FW2) | huNLRC4 C12 complex (EMDB-29498, PDB 8FW9) | Needle/huNAIP/huNLRC4 (R288A) (EMDB-29493, PDB 8FVU) |
|---|---|---|---|
| **Data collection and processing** | | | |
| Magnification | ×150,000 | ×150,000 | ×150,000 |
| Voltage (kV) | 200 | 200 | 200 |
| Electron exposure (e⁻ Å⁻²) | 43.8 | 43.8 | 43.8 |
| Defocus range (µm) | −0.8 to −2.4 | −0.8 to −2.4 | −0.8 to −2.4 |
| Pixel size (Å) | 0.948 | 0.948 | 0.948 |
| Symmetry imposed | C11 (Global) C1 (Focus) | C11 (Global) C1 (Focus) | C1 |
| Initial particle images (no.) | 128,266 | 128,266 | 869,012 |
| Final particle images (no.) | 72,822 | 21,406 | 106,381 |
| Map resolution (Å) | 3.80 | 4.46 | 3.60 |
| FSC threshold | 0.143 | 0.143 | 0.143 |
| Map resolution range (Å) | 3.5–8.0 | 4.4–9.0 | 3.3–6.0 |
| **Refinement** | | | |
| Initial model used (PDB code) | SWISS-MODEL | SWISS-MODEL | SWISS-MODEL |
| Model resolution (Å) | 4.0 | 5.0 | 3.7 |
| FSC threshold | 0.5 | 0.5 | 0.5 |
| Model resolution range (Å) | 4.0–6.0 | 5.0–7.0 | 3.7–6.0 |
| Map sharpening $B$ factor (Å²) | −106.4 | −148.4 | −95.4 |
| Model composition | | | |
| Non-hydrogen atoms | 2,700 | 2,700 | 2,334 |
| Protein residues | | | ATP,1 |
| Ligands | | | |
| $B$ factors (Å²) | | | |
| Protein | 80.9 | 213.1 | 68.3 |
| Ligand | | | 26.9 |
| r.m.s.d. | | | |
| Bond lengths (Å) | 0.0109 | 0.0114 | 0.0113 |
| Bond angles (°) | 0.96 | 0.98 | 1.01 |
| Validation | | | |
| MolProbity score | 1.26 | 1.30 | 1.49 |
| Clashscore | 1.31 | 1.56 | 2.21 |
| Poor rotamers (%) | 0.17% | 0.08% | 0.00% |
| Ramachandran plot | | | |
| Favored (%) | 93.96% | 94.07% | 91.73% |
| Allowed (%) | 5.30% | 5.07% | 6.86% |
| Disallowed (%) | 0.74% | 0.85% | 1.41% |

particles were also visible (Extended Data Fig. 2a). With a total of 108,418 cryo-EM images, two stable classes of disk-like particles were isolated and further refined by imposing C11, or C12 symmetry, respectively (Fig. 1c, Extended Data Figs. 1–3 and Table 1). NACHT and LRR domains formed the inner and outer rings in the Needle/huNAIP/huNLRC4 inflammasome (Fig. 1c) similarly observed previously in mNLRC4$_{ΔCARD}$ inflammasomes[16,17]. In addition, we identified the density corresponding to the CARD domains was protruding from the central disk of the huNLRC4 inflammasome (Fig. 1c,d). The CARD density was relatively poor compared to the NACHT and LRR domains. Isolated CARD domains adopted a helical structure that does not follow the C11 or C12 symmetry that we applied during the disk reconstruction[18]. With the 11-mer particles, the CARD domain density was slightly improved

from the refinement without imposing symmetry (Fig. 1d). By adjusting the threshold of the density map, the CARD was connected to the NACHT via multiple loops. A short segment of CARD helical structure can be docked into this density (Extended Data Fig. 4). Interestingly, the direction of this CARD helix is ~9° to the axis that is perpendicular to the disk plane (Fig. 1d). This tilted angle was driven by the tension from the linkers that connect the helical CARD domains to the C11 or C12 NACHT–LRR disk plane (Extended Data Fig. 4).

## NLRC4/NLRC4 interaction in the human inflammasome complex

The C11 and C12 particles were subjected to symmetry expansion and followed by local refinements focusing on three successive protomers

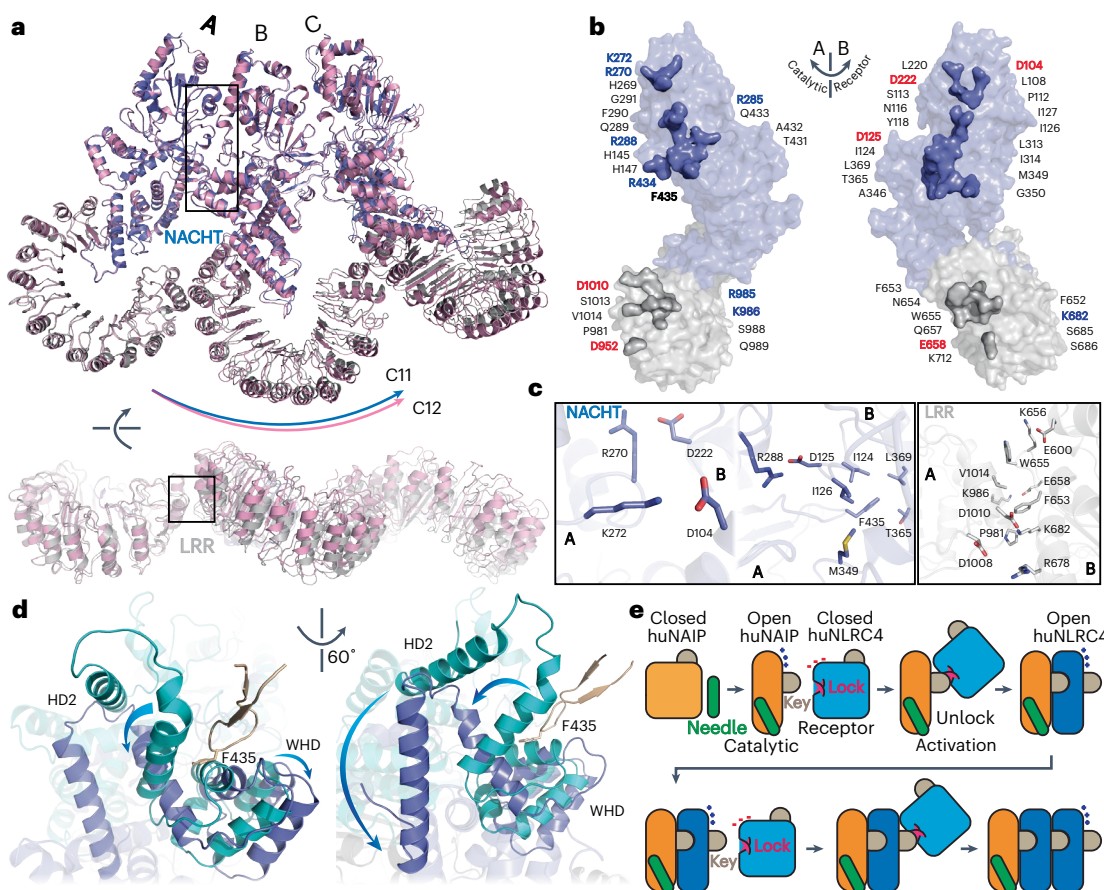

**Fig. 2 | Characterization of huNLRC4/huNLRC4 interactions. a**, Overlay of 12- and 11-mer tri-protomer structures; the individual protomers are labeled as A, B and C. The 11-mer and 12-mer tri-protomers are colored by domain organization and in pink, respectively. **b**, Surface representation with highlighted interaction residues of huNLRC4–huNLRC4 interfaces in the tri-protomer. The surface interfaces are colored to match their respective counterparts. The charged and 'key' residues are highlighted and colored on the basis of their properties: blue for positive, red for negative, and yellow for hydrophobic. **c**, Magnified views of the interfaces with main and side chains in cartoon and sticks, respectively. **d**, Structure overlay of the open (slate) and modeled closed (cyan) huNLRC4 at the 'lock–key' region. Wheat, 'key' from the catalytic open huNLRC4. The side chain of Phe$_{435}$ is shown as sticks. Subdomain reorganization upon activation is highlighted by arrows. **e**, Schematic representation of proposed 'lock–key' unlocking mechanism at the NACHT domain. LRR is not shown.

(tri-protomer) (Extended Data Fig. 2). Both tri-protomers were modeled as only containing huNLRC4. The structural overlay of these tri-protomers at Protomer$_{PositionA}$ shows outward assembly of the C12 tri-protomer relative to the C11 tri-protomer (Fig. 2a). This outward assembly is accompanied by the slight tilt of LRR relative to the NACHT, implying plasticity of the activated huNLRC4. Both tri-protomers share conserved structures and protomer-protomer interfaces. For the comparison of activated huNLRC4 and mNLRC4$_{ΔCARD}$, we used the C11 tri-protomer, representing the Needle/huNAIP/huNLRC4 inflammasome structure.

The activated huNLRC4 and mNLRC4$_{ΔCARD}$ structures are similar with root-mean-square deviation (r.m.s.d.) 2.38 Å. The huNLRC4/huN-LRC4 interfaces contain several conserved residues and intermolecular salt bridge pairs (Fig. 2b,c and Extended Data Fig. 5a). Compared to mNLRC4, huNLRC4 has two additional salt bridge pairs (Asp104–Lys272 and Asp222–Arg270), while both the Asp at positions 104 and 222 are Asn residues in mNLRC4. Noteworthily, the Phe435 from the activated protomer is positioned in a well-defined hydrophobic pocket in the receptor protomer (Fig. 2c,d). This pocket contains several conserved residues (Ile124, Ile126, Ala346, Met349, Thr365 and Leu369) from both the NBD and WHD subdomains undergoing large conformational changes during activation (Fig. 2d). Interestingly, this pocket exists only in the activated open huNLRC4 and is partially formed in the closed model (Fig. 2d). Insertion of Phe435 into this partially formed pocket may unlock the closed huNLRC4, triggering the conformational

changes required for subsequent oligomer assembly (Fig. 2e). In mNLRC4, this residue is Leu435 (Extended Data Fig. 5b). However, mNLRC4-L435D mutation still displayed flagellin-induced interaction with NAIP5 but failed to form higher-order oligomeric complex[16]. Here we propose a 'lock–key' mechanism for propagating huNLRC4 self-assembly into a full ring-like structure (Fig. 2e and Extended Data Fig. 5b). The charge-complemented surfaces of the NBD–NBD bring the catalytic 'key' in proximity to the receptor 'lock' surface, and the subsequent insertion of this 'key' induces a conformational change in NACHT. The LRR–LRR interface provides an additional docking site, rigidifying the activated conformation to unlock the subsequent huNLRC4s. Sequence alignment further indicates huNLRC4 has both the receptor 'lock' for its own activation and the catalytic 'key' for donating to activate the closed huNLRC4, while huNAIP only has the catalytic 'key' without a receptor 'lock' site (Extended Data Fig. 5b and Supplementary Information). Hence, open huNAIP can activate closed huNLRC4, but neither open huNAIP nor huNLRC4 can directly activate closed huNAIP. Indeed, a trigger such as T3SS Needle is required for huNAIP activation and then huNLRC4 priming (Fig. 2e).

## Cryo-EM structure of the Needle/huNAIP/huNLRC4-R88A complex

Unlike NLRC4, NAIP hosts three baculovirus IAP-repeat (BIR) domains (Fig. 1a). However, multiple rounds of 3D classifications without

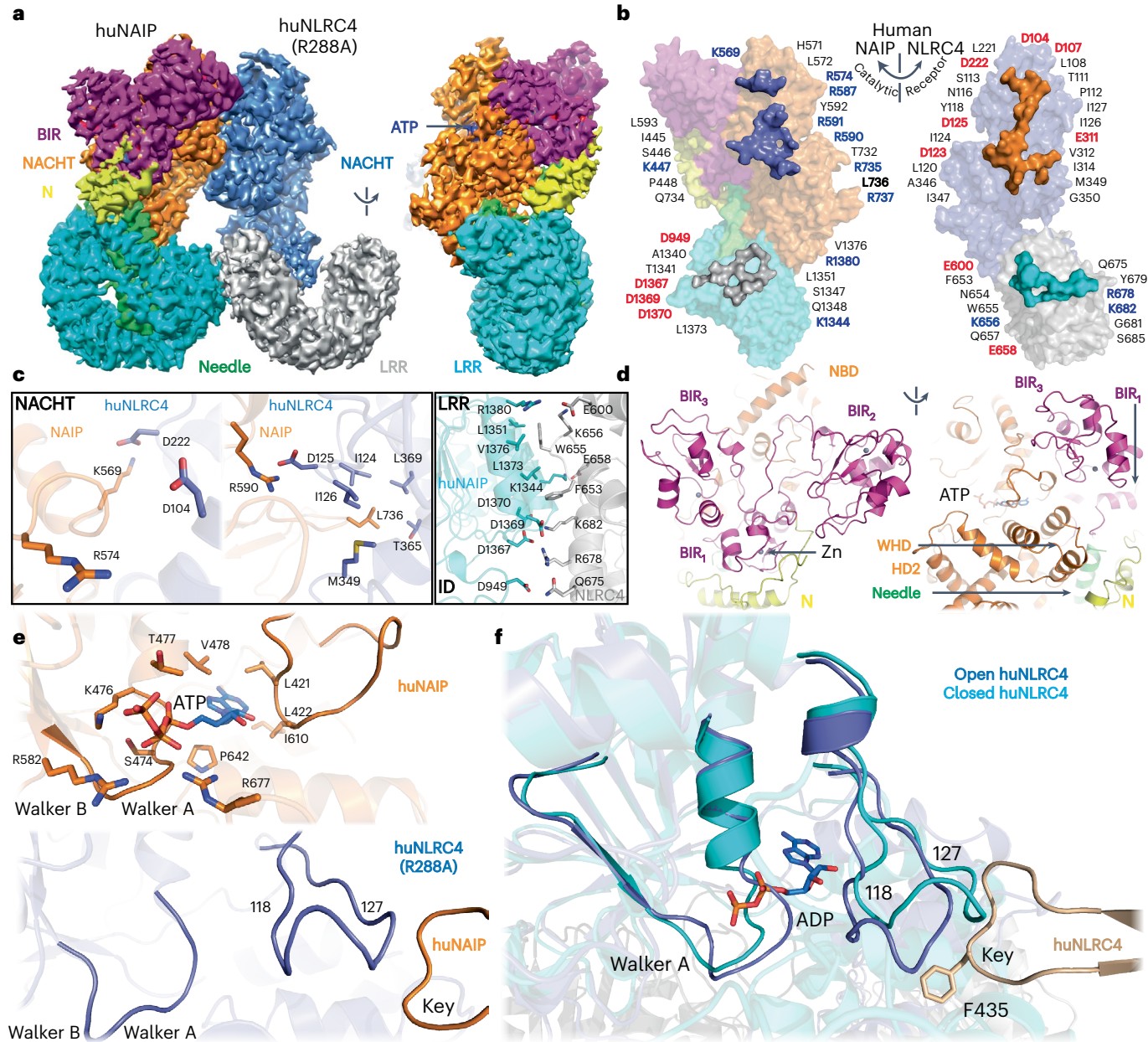

**Fig. 3 | Structure of the Needle/huNAIP/huNLRC4-R288A tripartite complex.**
**a**, Cryo-EM map of Needle/huNAIP/huNLRC4-R288A Maps are colored by domain organization. **b**, Surface representation with highlighted interaction residues of huNAIP–huNLRC4 interfaces. The surface interfaces are colored to match their respective counterparts. The charged and 'key' residues are highlighted and colored on the basis of their properties: blue for positive, red for negative and yellow for hydrophobic. **c**, Magnified views of the interfaces with main and side chains in cartoon and sticks, respectively. **d**, Zoom-in views of huNAIP BIR domains. **e**, Magnified views of nucleotide-binding pockets in the open human NAIP (top) and NLRC4-R288A (bottom). Main and side chains of residues around ATP are shown as cartoon and sticks, respectively. **f**, Structure overlay of the open and modeled closed huNLRC4 at the nucleotide binding site. Wheat, 'key' from the catalytic open huNLRC4. The side chain of Phe435 and ADP is shown as sticks. Loop$_{118-127}$ is highlighted.

alignment did not separate any particle with features from NAIP–BIRs nor Needle. The symmetry operations that we applied during the structural reconstruction of the WT disk may average out the features. The huNLRC4-R288A mutant preferably forms a complex composed of NeedleTox, huNAIP and huNLRC4-R288A in equal molar ratios (Fig. 1b). Cryo-EM analysis of the intermediate Needle/huNAIP/huNLRC4-R288A complex, without symmetry operation, revealed a structure with two apparent NLRC4-like protomers (Fig. 3a, Table 1 and Extended Data Figs. 2 and 3). The quality of the EM density map allowed us to model one protomer as huNAIP with features of BIRs which embraces Needle, and the other as an activated huNLRC4-R288A interacting with

huNAIP forming extensive charged, polar and hydrophobic interactions (Fig. 3b,c).

## NAIP/NLRC4 interactions in priming the NLRC4 activation
The huNAIP 'Key', Leu736, interacts with the 'lock' site in the activated huNLRC4-R288A (Fig. 3c and Extended Data Fig. 5b). Similar to the huNLRC4 tri-protomer at the NACHT interfaces, three pairs of salt bridge bonds were formed between huNAIP Lys569, Arg574 and Arg590 and huNLRC4-R288A Asp222, Asp104 and Asp125, respectively (Fig. 3c). In contrast, there are two additional salt bridges (Arg1380–Glu600 and Asp1367–Arg678) between the huNAIP/huNLRC4-R288A

LRRs. huNAIP hosts one short amino-terminal domain (N) followed by three tandem BIR domains (BIR$_{1-3}$) (Fig. 1a). The BIR$_{1-3}$ abutting the NACHT domain forms extensive interactions with the NBD and WHD subdomains, while the N protruding toward the LRR interacts with both WHD and HD2 subdomains (Fig. 3a,d). There are no direct contacts between the BIR$_{1-3}$ and huNLRC4-R288A. Each BIR harbors one zinc-finger motif including three cysteines and one histidine, which coordinates a zinc ion. Relatively strong densities were observed for each zinc ion binding site suggesting that the zinc ions were copurified with the protein (Extended Data Fig. 3). The loop$_{352-393}$ connecting the BIR$_3$ and NACHT was not modeled due to the poor density at that region, indicating a dynamic character. A non-protein density located in the core of NACHT was modeled as ATP, forming extensive interactions with the Walker A motif, and WHD subdomain (Fig. 3e and Extended Data Fig. 3). Three positively charged residues (NBD: Lys476 and Arg582; WHD: Arg677) surround the triphosphate tail, while Arg582 is a histidine residue in the murine NAIPs (Fig. 3e top and Supplementary Information). Unlike the closed mNLRC4[19], no nucleotides were observed in the open huNLRC4 despite huNAIP and huNLRC4 sharing a similar nucleotide binding motif (Fig. 3e bottom, Extended Data Fig. 5a and Supplementary Information). Insertion of the 'key' from either huNAIP or huNLRC4 narrows the nucleotide-binding pocket via loop$_{118-127}$ in the receptor huNLRC4, making it unfavorable for nucleotide binding (Fig. 3e,f). In contrast, huNAIP does not possess the receptor 'lock' site for this 'key' insertion; hence, its nucleotide-binding pocket is preserved in its open form (Fig. 3e and Extended Data Fig. 5b). The distinct nucleotide binding modes imply that the nucleotide binding may play distinct roles between huNAIP and huNLRC4[20,21].

The huNAIP has a unique feature that the last eight residues pass through the insertion domain (ID), and pull the ID toward the huNLRC4–LRR, creating additional hydrogen-bond interactions between huNAIP Asp949 and huNLRC4 Gln675 and Arg678 (Figs. 1a, 3b,c and 4a). We call this region the 'lasso-like motif' because of its shape. The lasso-like motif was also observed in the open murine NAIP5 structure, thereby indicating it is a conserved element across all murine NAIPs and huNAIP (Supplementary Information)[14].

### Pathogen sensing through Needle/huNAIP interactions

We have identified here that the bacterial T3SS Needle triggers the activation of huNAIP initiating the Needle/huNAIP/huNLRC4 inflammasome complex assembly. In the structure, Needle adopts a helix–hairpin–helix structure that forms extensive interactions with huNAIP (Fig. 4a and Extended Data Fig. 6a). Needle C-terminus was anchored into a narrow pocket formed by the huNAIP N, BIR$_1$ and HD1 domains. Needle Arg375, Lys364, and Asp367 form salt bridge interactions with N Asp16, Glu37 and HD1–Lys631, respectively. Needle Val371 and Phe374 were in a hydrophobic pocket composed of N Leu20,23,27,34,38 and BIR$_1$–Leu109, and HD1 Met626 and Val627. Needle N-terminus was positioned to one side of a wide opened cleft formed by huNAIP N, NACHT and ID domains with hydrophobic interactions between Needle Trp300 and huNAIP Val865, Leu868 and Tyr872, methylene groups of Lys869, and Leu900. Several intra- and inter-hydrophobic interactions were formed within Needle's two-helix bundle and between the bundle and the channel formed by ID, the lasso-like motif and LRR. One salt bridge (Glu348–Arg1358) was formed between Needle and huNAIP LRR. It has been reported that human and murine NAIPs respond to Needle, flagellin or PrgJ, but not to SsaI[7,8,10,11]. A hydrophobic residue followed by a conserved Arg residue at the C-terminus plays an important role in the recognition, while this pattern is missing in SsaI (Fig. 4a and Extended Data Fig. 6b). In addition, Needle, Flagellin and PrgJ can adopt a helix–hairpin–helix structure with similar surface charge distributions, whereas the Alphafold2[22] predicted SsaI model prefers an alternative extended conformation (Extended Data Fig. 6c).

### Structural basis of huNAIP activation

A structural overlay of the activated huNAIP and mNAIP5[14] showed that mNAIP5 has a deeper lasso-like motif with five more a.a. passing through the ID loop (Fig. 4b,c). The variable tightness of the lasso-like motif results in the closure of the horseshoe-like structure of LRR with different diameters at the center. To investigate the impact of the lasso-like motif on the dynamics of huNAIP, we performed molecular dynamics (MD) simulations on the WT and a carboxy-terminal truncate where the nine threaded residues are deleted. In the WT simulation, the lasso-like motif was maintained throughout the simulation. To compare the overall flexibility, root mean squared fluctuation of all α-carbon atoms were computed (Fig. 4d and Extended Data Fig. 7) after aligning the trajectories on the cryo-EM structure using the backbone of the NACHT domain. The C-terminal truncate simulation that lacks the lasso-like motif shows increased fluctuations for the whole protein, including the NACHT domain on which the trajectories were aligned. Notably, the ID domain that forms the loop for the lasso also shows greater fluctuations. The simulations suggest that the lasso-like motif rigidifies NAIP, and this rigidification may play a role in inflammasome assembly. A Des 9 C-terminal truncation on huNAIP (huNAIP$_{1-1394}$) was introduced to investigate this hypothesis and assess the impact of the lasso-like motif on the human NLRC4 inflammasome assembly. Purified NeedleTox, FL huNAIP or huNAIP$_{1-1394}$, and huNLRC4 were mixed in a 1:1:10 molecular ratio and then subjected to analytical gel filtration analysis. Compared to the FL huNAIP, huNAIP$_{1-1394}$ still displayed Needle-induced interaction with huNLRC4 but with a compromised HMW oligomeric complex, accompanied with the free huNLRC4 monomers (Fig. 4e). Since the truncated sequence does not directly interact with the huNLRC4, perturbing the lasso-like motif will destabilize the Needle/huNAIP binary complex. These observations indicate that a rigid lasso-like motif is required for human NAIP/NLRC4 inflammasome assembly.

The adjustable tightness of the lasso-like motif assists in the activation of huNAIP (Fig. 4f). We speculate that this motif forms upon Needle binding to huNAIP, since a preformed lasso-like motif would limit trigger recognition (Fig. 4a,b). Upon binding, the ID and lasso-like motif exert a long-range effect on the NACHT domain via the Needle's C-terminal helix pushing into the pocket formed by huNAIP N, BIR$_1$ and HD2 domains, thereby activating huNAIP (Fig. 4f). Indeed, the C-terminal 35-residue peptide from flagellin is sufficient to activate mNLRC4-mediated immune response[23]. From the structure, the Needle's C-terminal helix bridges the lasso-like motif, ID, and the pocket with extensive interactions (Fig. 4a). This lasso-like motif appears to function as a 'cable tie' strengthening the Needle/huNAIP complex to activate huNAIP and lock it in an open conformation (Fig. 4f).

### Proposed huNLRC4 oligomeric assembly mechanism

The last step of inflammasome assembly is the incorporation of the 10th or 11th huNLRC4, ending with a fully assembled inflammasome disk. The structures of the Needle/huNAIP/huNLRC4-R288A and the huNLRC4$_{PositionA}$/huNLRC4$_{PositionB}$/huNLRC4$_{PositionC}$ tri-protomer allowed us to extrapolate the mechanism of this final step. Both complexes were super-imposed using the huNLRC4-R288A and huNLRC4$_{PositionC}$ (Fig. 5a). huNLRC4$_{PositionB}$ and huNLRC4$_{PositionA}$ may represent the huNAIP in the fully assembled disk, and the last huNLRC4 closing the disk, respectively. Structures of huNLRC4-R288A and huNLRC4$_{PositionC}$ can be overlaid with r.m.s.d. 2.1 Å, while major clashes were observed between the huNAIP BIR$_3$ and huNLRC4$_{PositionA}$–NACHT (Fig. 5b,c middle). Both BIR$_{1-2}$ would need to adjust their positions since the loops connecting the BIRs would clash with the huNLRC4$_{PositionB}$–NACHT (Fig. 5b,c right). Notably, the huNAIP LRR would need to rotate ~20° toward the Needle to be incorporated into the disk (Fig. 5a,c left). This structural comparison highlighted the major structural rearrangements needed on BIR$_{1-3}$ and LRR for huNAIP to be incorporated into the fully assembled disk. These rearrangements would either cause the release of Needle from huNAIP or the entire Needle/huNAIP complex

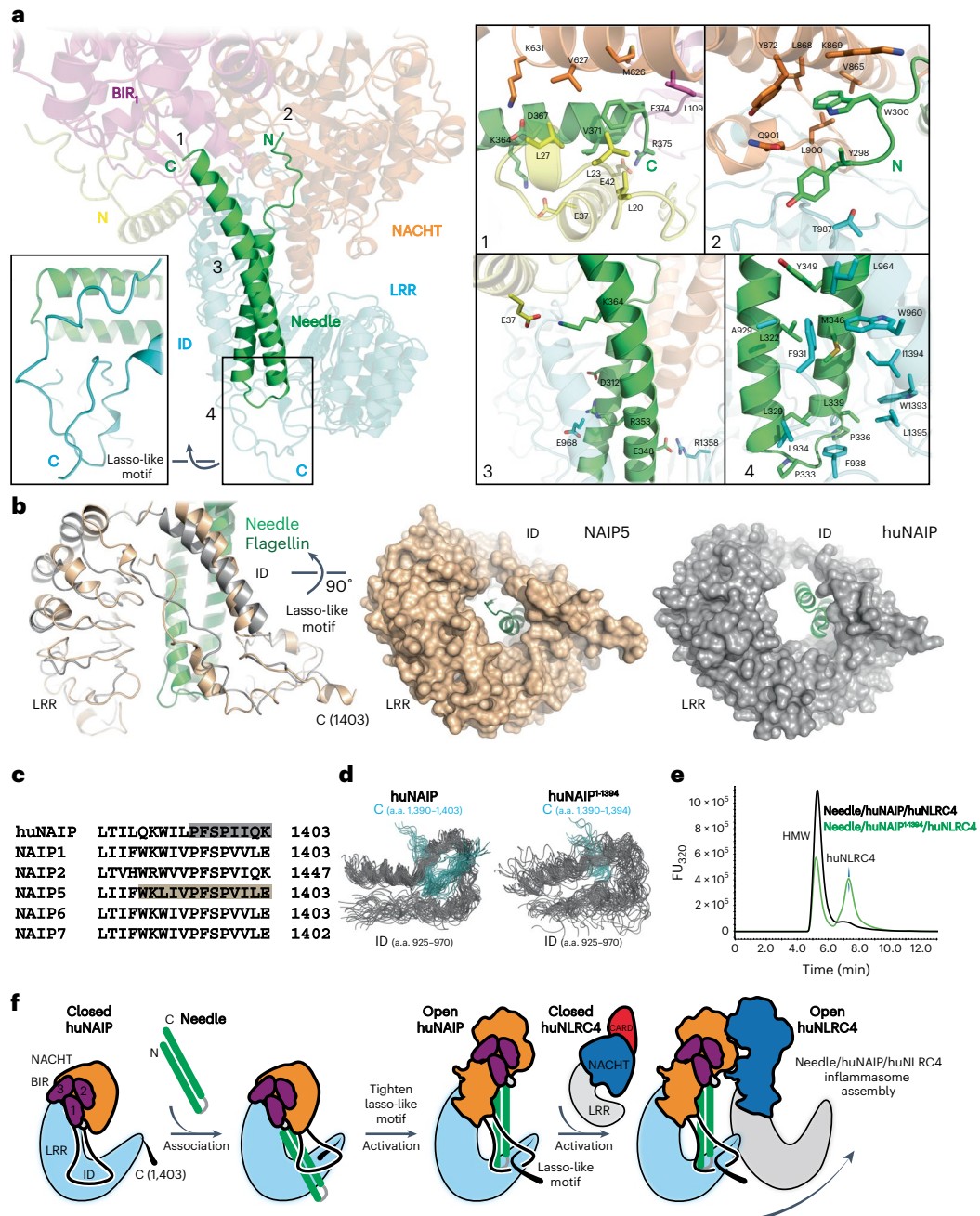

**Fig. 4 | Characterization of Needle/huNAIP interactions. a**, Expansion of Needle/huNAIP interaction with magnified views of four interaction regions (1–4) and lasso-like motif. Main chains, side chains, ATP and Zn ions are shown as cartoon, sticks, sticks and spheres, respectively, and colored by domain organization. **b**, Left: structural overlay of huNAIP and NAIP5 at the lasso-like motif. HuNAIP, NAIP5, flagellin and Needle are shown in cartoon and colored in gray, wheat, dark green and light green, respectively. Right: the LRRs are shown as surface highlighting the diameter difference between huNAIP and NAIP5. Flagellin and Needle are shown in cartoon and colored in dark green and light green, respectively. **c**, Sequence alignment at the C-terminus of NAIPs. The sequences passing through the ID loop in huNAIP and NAIP5 are highlighted in gray and wheat, respectively. **d**, Expansion of the 50 frames from an MD simulation of huNAIP394–1403 (left) and huNAIP394–1394 (right) truncated at the C-terminus. The trajectories are aligned on the loop resides forming the lasso-like structure. **e**, Analytical gel filtration analysis of purified huNLRC4 inflammasome with and without C-terminal truncation. FU, fluorescence units. **f**, Schematic representation of proposed huNAIP activation steps. The Apo NAIP adopts a dynamic conformation without formation of the lasso-like motif. In its open form, NAIP creates a large association interface to rapidly capture the Needle. The engagement of Needle guides the C-terminal region through the ID loop, tightening the lasso-like motif and strengthening the complex. This activated NAIP/Needle complex provides a 'key' (Leu736) and a large catalytic interface (Fig. 3b,c) for the activation of closed huNLRC4, initiating the assembly of the human NLRC4 inflammasome.

from the disk (Fig. 5d models 1 and 2). Both hypotheses are consistent with the observation that there are no Needle nor BIR$_{1–3}$ features in the assembled disk (Fig. 1c). The released Needle or Needle/huNAIP complex may initiate another round of inflammasome assembly or be subject to degradation to stop further immune activation, depending on the insult. Interestingly, stacked disks with multi-layers from the flagellin/mNAIP5/mNLRC4 inflammasome were reported previously[24]. We observed similar stacked Needle/huNAIP/huNLRC4 inflammasome particles in raw images (Extended Data Fig. 2b). This implies that, instead of a large structural rearrangement of huNAIP, huNLRC4 may

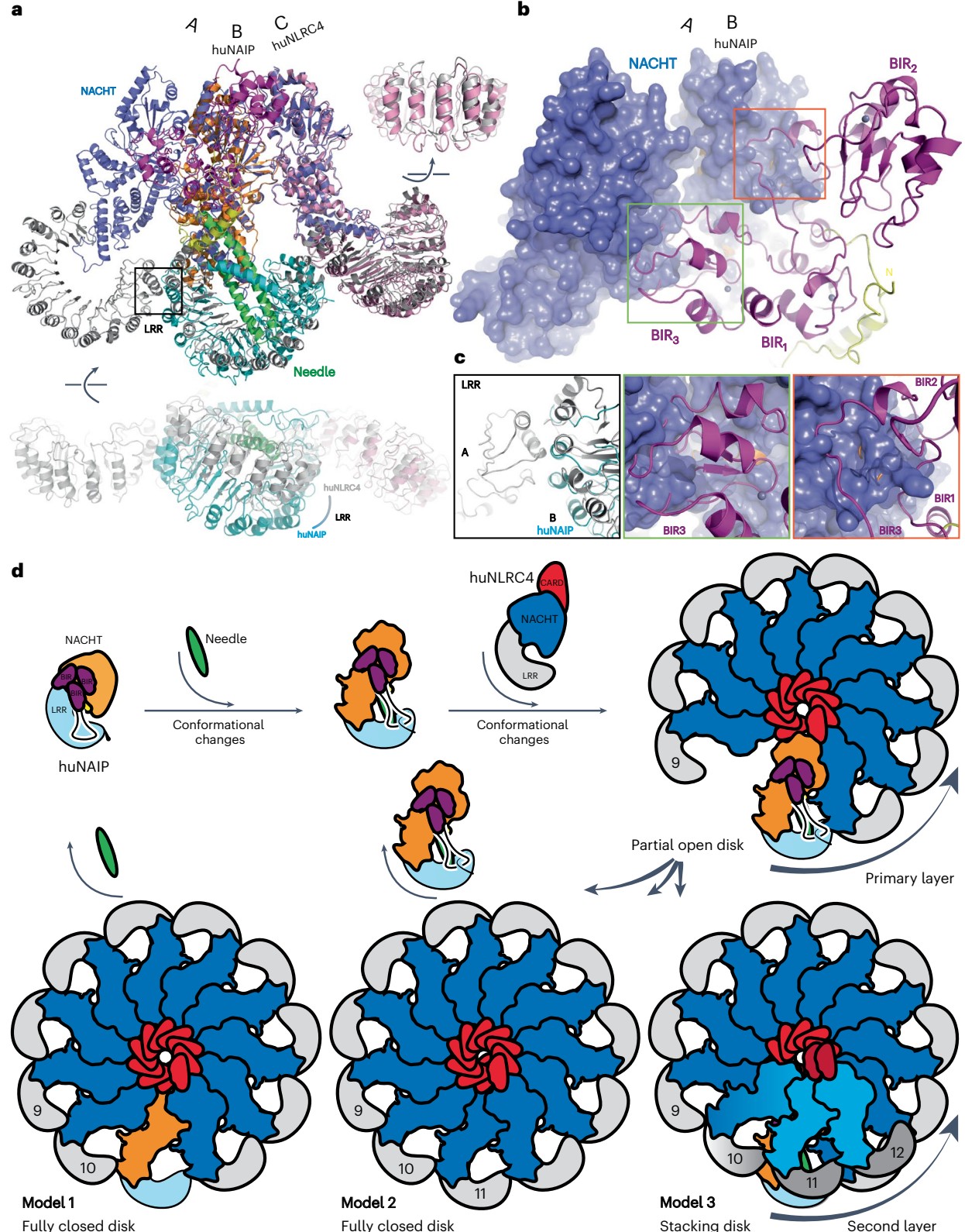

**Fig. 5 | Purposed model of the full huNLRC4 inflammasome assembly.**
**a**, Overlay of Needle/huNAIP/huNLRC4-R288A and tri-protomer structures. Structures are colored by domain organization with huNLRC4-R288A in pink. **b**, Zoom-in view of NACHT domains on position AB. NACHT domains from tri-protomer and huNAIP are shown as surface and cartoon, respectively. **c**, Magnified views of three interaction regions highlighted in **a** and **b**. **d**, Proposed huNLRC4 inflammasome assembly model: Needle binding activates huNAIP, forming a stable Needle/huNAIP binary complex. The activated huNAIP

then activates closed huNLRC4, initiating huNLRC4 inflammasome assembly (Fig. 4f). In the final step as shown, the tenth huNLRC4 incorporates into the disk, interacting with NAIP from the backside. Conformational changes in NAIP are required to fully close the assembly disk, which may result in the release of the Needle from the Needle/NAIP complex (model 1), the release of the Needle/NAIP complex from the disk (model 2) or continuous stacking of NLRC4 on top of the Needle/NAIP complex to form a stacking disk (model 3).

continue to assemble on top of the Needle/huNAIP, forming spirally stacked disks (Fig. 5d model 3). These stacked disks, only observed from the FL murine[24] or huNLRC4 inflammasomes, are possible due to the central CARD helical structure along with the plasticity of NLRC4-LRR, the bulky BIR$_{1-3}$ of NAIPs, and the rigid Needle/huNAIP complex, working synergistically. Nevertheless, we cannot exclude that this might be a result of our in vitro system. Future studies within a cellular context could validate this assembly process.

## Discussion

The cryo-EM structures of the FL NAIP/NLRC4 complex provide critical structural insights into the formation of the NAIP/NLRC4 inflammasome in humans. Our results shed light on the precise molecular modalities of pathogen sensing by huNAIP to instigate an innate immune response through NLRC4. From the structures, huNAIP or mNAIP5 form interactions with more than half of the sequence of Needle or Flagellin, respectively, in which 50% of interactions are hydrophobic (Extended Data Fig. 6a). A large ratio of hydrophobic to hydrophilic interactions may accelerate the association rate between the NAIP and trigger, but with modest specificity, which enhances sequence tolerance, meanwhile reducing the risk of escape mutations. Indeed, the recognition of mNAIP5 to Flagellin was not abolished by point mutations[14,15]. In addition, the adjustable tightness of the lasso-like motif may grant NAIP next level of freedom in the trigger recognition since the recognition interface may vary accompanying the tightness of the lasso-like motif (Fig. 4f). Unlike murine NAIPs, the single huNAIP isoform has been reported to broadly respond to Needle, Flagellin and PrgJ in human macrophages[2,3,25]. Here we show that NeedleTox but neither RodTox nor FlaTox is sufficient to induce a robust and complete inflammasome formation in our biochemical system (Extended Data Fig. 1). Our biochemical assays demonstrate the formation of a stable complex between Needle and huNAIP, enabling structural studies. However, we cannot disregard the possibility that weaker binding triggers such as RodTox or FlaTox might also be sufficient to activate NLRC4 inflammasome assembly in human macrophages. Furthermore, broad recognition was observed in human primary cells but not in THP-1 and U937 cell lines, possibly due to the presence of the huNAIP isoform specifically expressed in primary cells[26]. Moreover, post-translational modifications (such as phosphorylation and potentially ubiquitinition) in human cell types may differ from the mouse system[27,28]. These modifications and their interactions with other regulators could enhance the broad recognition of human NAIP toward various triggers. Currently, we only have interaction profiles available for mNAIP5/Flagellin and huNAIP/Needle[15]. Consequently, we are unable to fully explain why human NAIP recognizes multiple triggers solely on the basis of the available structural information. Further structural studies involving both murine and human NAIPs in conjunction with other triggers could shed light on the selection mechanisms for pathogen sensing.

The human NAIP and NLRC4 apply distinctive activation mechanisms. The recent structure of mNAIP5 alone helps to get a sense of NAIP's plasticity[21]. Unlike NLRC4, NAIP alone does not possess a receptor surface. The association with the triggers followed by the tightening of the lasso-like motif synergistically activates the NAIP (Fig. 4f). From the structure of Needle/huNAIP/huNLRC4-R288A, the C-terminus of the Needle forms direct interactions with huNAIP BIR$_1$ domain that may rigidify the other BIRs. The rigidified BIR$_{1-3}$ and the trigger can serve as the fiducial markers for locating the NAIP in the NLRC4 inflammasome disk structures. NAIP was captured in a partially assembled flagellin/mNAIP5/mNLRC4 open disk[15]. However, NAIP was not observed in fully assembled closed disk structures of both murine and human NLRC4 inflammasomes[17]. Our structural overlays further indicated the BIR$_3$ may clash with the last NLRC4 within the fully assembled disk and large conformational changes are needed for Needle/huNAIP to be fully incorporated into the assembled disk, which may induce the release of the Needle and rearrangement of the BIR$_{1-3}$ domains (Fig. 5d).

Furthermore, using analytical size exclusion column and mass photometry Refeyn, we observed a notable peak that may correspond to the free NAIP or Needle/NAIP when mixed with WT NLRC4 (Extended Data Fig. 1g,h). A technique such as cryogenic electron tomography may be helpful to accurately visualize the inflammasome and determine if the Needle/NAIP complex is present in the fully assembled closed disk.

A short segment of CARD/PYRIN domain helical structure positions at the center of the activated huNLRC4/NLRP3 disk structures, respectively (Fig. 1d and Extended Data Fig. 4)[29]. Green fluorescent protein (GFP) fusion and point mutations ($F_{79}A$ and $D_{83}A$) in the mNLRC4 CARD domain resulted in a partially assembled flagellin/mNAIP5/mNLRC4 open disk. Our unpublished data using N-Term SumoStar tagged NLRC4, a smaller domain than GFP, showed a partially open disk. Removing the N-term tag, with or without the CARD, resulted in the fully assembled disk. These findings suggest that the N-terminal-tagged NLRC4 CARD sterically hinders NLRC4 inflammasome assembly[15,17]. However, the CARD is dispensable for the NLRC4 activation and disk formation[17], while the presence of PYRIN domain, ASC, and reorganization of oligomerization state play critical roles in the NLRP3 activation and disk formation[29]. The striking differences of oligomerization states, and PYRIN/CARD domain requirements in the disk assembly, indicate that NLRP3 and NLRC4 use distinctive pathways in their activation. We proposed a structural mechanism critically involved in the regulation of disk assembly through 'lock–key' interactions in NLRC4 (Fig. 2e and Extended Data Fig. 8). Activated NAIP or NLRC4 donates a large catalytic surface hosting one 'key' at the NACHT for NLRC4 activation. The LRR provides a secondary docking site for the NLRC4 activation that is missing in the NLRP3 inflammasome structure[29]. The activated NLRP3 disk structures showed that centrosomal kinase NEK7 does not directly participate in NLRP3 oligomerization.

Since nucleotides were not present in the open NLRC4 structures, we speculate the NLRC4 activation does not require ATP hydrolysis. This is consistent with the low ATP hydrolysis activity observed with the FL huNLRC4 in our reconstituted system. We hypothesize that the ADP observed bound in the closed mNLRC4 monomers[19] plays more of a structural function and the 'lock–key' insertion mechanism facilitates ADP release upon the complete opening of huNLRC4 (Fig. 3e,f). In line with that, most NLRC4 gain-of-function mutations are close to the nucleotide binding site and thereby might subsequently reduce ADP affinity and perturb the protein stability. This event might be sufficient to open and hyperactivate huNLRC4 to trigger inflammatory diseases. This lack of dependency of huNLRC4 inflammasome assembly on ATP hydrolysis contrasts with other inflammasomes including NLRP3 and NLRP1[30–32].

The NAIP/NLRC4 inflammasome in humans is crucial to the host immune response to bacterial infections and contributes to auto-immune and inflammatory diseases in a sterile condition. These structural insights presented here pave the way to tailored development of important specific anti-infectious and anti-inflammatory therapeutic agents.

## Online content

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

## Methods

### Recombinant proteins

All gene synthesis and cloning was done at Epoch Life Science. The optimized DNA sequences are listed in Supplementary Data 1 and inserted into the vectors described. Recombinant flagellin of *Legionella pneumophilia* with an N-terminal lethal factor (Lfn) known as FlaTox was purchased from VIB Protein Service Facility.

Purifications were done at 4 °C unless otherwise specified.

6His.Lfn.Needle of *Bacillus thailandensis* (NeedleTox) and 6His.Lfn.PrgJ of *Salmonella* Typhimurium (RodTox) were generated as previously described[33] with modifications in the purification. NeedleTox was released from collected cell paste by pressure drop lysis through a microfluidizer in 25 mM HEPES pH 7.5, 0.5 M NaCl. NeedleTox was captured by HisTrap HP column and eluted with 0.4 M imidazole before purification Superdex 200 with phosphate-buffered saline at pH 7.4. RodTox was released from collected cell paste by pressure drop lysis through a microfluidizer in 25 mM HEPES pH 7.5, 0.3 M NaCl with 1× protease inhibitors and purified with Ni-NTA resin followed by heparin and then polished on Superdex 200 with phosphate-buffered saline at pH 7.4.

DNA encoding FL human NLRC4 (UniProt Q9NPP4) and FL human NLRC4 R288A were engineered with FLAG.8His.SUMOstar.CfaC[34] at the N-terminus and inserted into a PVL1393 vector (Epoch Life Sciences). P2 BIICs were generated and used to infect Sf9s for 96 h at 27 °C. NLRC4 WT and R288A mutant were released from cell paste by sonication in 25 mM Tris–HCl pH 8.0, 150 mM NaCl, 0.2 mM tris(2-carboxyethyl) phosphine (TCEP) and 20% glycerol with Complete EDTA protease inhibitors (Sigma-Aldrich). NLRC4 constructs were captured from clarified lysate supernatant by His-Pur Ni-NTA resin (Thermo Fisher Scientific) in the presence of 25 mM imidazole. The Ni resins were washed with 2 M NaCl followed by a 10 mM ATP–MgCl$_2$ wash in lysis buffer and then eluted with 0.4 M imidazole. The Ni eluates were further purified with anti-DYKDDDDK resin (Pierce) and NLRC4 constructs were eluted with 0.25 mg ml$^{-1}$ H2N-DYDDDDK-OH (New England Peptide) in lysis buffer. The N-terminal tags were cleaved by overnight incubation with CfaN peptide[34]. The monomeric peak from Superdex 200 in 25 mM HEPES pH 7.4, 0.4 M NaCl, 0.2 mM TCEP and 20% glycerol was pooled.

DNA encoding FL human NAIP (UniProt Q13075) or C-terminal Des9 truncate (huNAIP$_{1-1394}$) were engineered with FLAG.8His.MBP.TEV.CfaC at the N-terminus and inserted into a PVL1393 vector (Epoch Life Sciences). P2 BIICs were generated and used to infect Sf9s for 96 h at 27 °C. FLAG.8His.MBP.TEV.CfaC.NAIP constructs were released from cell paste by sonication in 25 mM HEPES pH 7.4, 0.4 M NaCl, 0.4% CHAPS, 20% glycerol and 0.2 mM TCEP plus Complete EDTA protease inhibitors and captured from the clarified lysate supernatant onto amylose resin (NE BioLabs). After a 2 M NaCl and a 10 mM ATP–MgCl$_2$ wash the FLAG.8His.MBP.TEV.CfaC.NAIP constructs were eluted with 20 mM maltose in lysis buffer.

The amylose eluates were either loaded directly onto Superose 6, or the N-terminal tag was removed by incubation with the CfaN peptide. The monomeric peaks were pooled after polishing on Superose 6 with 25 mM HEPES pH 7.4, 0.4 M NaCl, 0.4% CHAPS, 0.2 mM TCEP and 20% glycerol.

### Analytical in vitro inflammasome formation

HuNLRC4, huNAIP, huNAIP$_{1-1394}$ and assorted triggers (FlaTox, NeedleTox or RodTox) were evaluated by Trp fluorescence on an analytical Superdex 200 5/150 in 25 mM HEPES pH 7.4, 0.2 M NaCl and 0.2 mM TCEP as either a mixture of 10:1:1 (NLRC4:NAIP:trigger) or individual subunits.

### In vitro inflammasome formation for cryo-EM

HuNLRC4, FLAG.8His.MBP.TEV.CfaC.NAIP and NeedleTox were complexed in final molar ratio of 10:1:1, respectively. HuNLRC4-R288A, huNAIP and NeedleTox were complexed respectively in a 10:1:1 molar ratio. Intact complexes were isolated on Superose 6 in 25 mM HEPES pH 7.4, 0.2 M NaCl and 0.2 mM TCEP.

### Grid preparation and data acquisition

A total of 3.5 μl of purified unconcentrated Needle/huNAIP/huNLRC4 or 0.2–0.3 mg ml$^{-1}$ Needle/huNAIP/NLRC4$_{R288A}$ complex was applied to the plasma-cleaned (Gatan Solarus) Quantifoil 1.2/1.3 UltraAuFoil holey gold grid, and subsequently vitrified using a Vitrobot Mark IV (FEI Company). Cryo grids were loaded into a Glacios transmission electron microscope (Thermo Fisher Scientific) operating in nano-probe at 200 keV with a Falcon IV direct electron detector. Images were recorded with Thermo Scientific EPU software in counting mode with a pixel size of 0.948 Å and a nominal defocus range of −2.4 to −1 μm. Data were collected with a dose rate of 5.3 electrons per physical pixel per second, and images were recorded with a 7.2 s exposure in electron-event representation format corresponding to a total dose of 43.8 electrons Å$^{-2}$. All details corresponding to individual datasets are summarized in Table 1.

### EM data processing

For the Needle/huNAIP/huNLRC4 complex, a total of 108,418 movies were collected from multiple datasets. Per dataset, the movies were subjected to beam-induced motion correction, contrast transfer function parameter estimation, automated reference particle picking, particle extraction with a box size of 480 pixels, and 2D classification in CryoSPARC[35] live during the data acquisition. Particle images with clear disk features were merged and subjected to ab initio 3D reconstruction with C11 symmetry in CryoSPARC. Multiple rounds of optimized 3D heterogeneous refinement yielded one class with a clear 11-mer density, containing a total of 72,822 particles. Three-dimensional reconstruction using the particles from the rest classes preserved the disk features, but the resolution was not improved imposing the C11 symmetry. The particles were further analyzed with 2D classification revealing a 12-mer feature and then subjected to ab initio 3D reconstruction with C12 symmetry. Three-dimensional heterogeneous refinement yielded one class with a clear 12-mer density, containing a total of 21,406 particles. The 11- or 12-mer particles were refined with C11 or C12 symmetry, respectively. Particles were then subjected to symmetry expansion, and then three successive protomers were masked and subjected to local refinement with C1 symmetry with the standard deviation of prior over rotation/shift parameters set to 3° and 2 Å, respectively. To improve the map quality of the CARD part, local focused refinement was carried out using the 11-mer particles without imposing symmetry. Thinking that the NACHT–LRR disk features may be impeding image alignment, signal subtraction to remove the signal from the NACHT–LRR disk was tried with a soft mask in the particle images. The signal-subtracted particles were directly input into refinement with/without helical symmetry but did not further improve the map quality of the CARD part. For the Needle/huNAIP/huNLRC4-R288A complex, all steps before cryoSPARC ab initio reconstruction were the same as for the Needle/huNAIP/huNLRC4, with the extraction box size of 420 pixels. Starting from the cryoSPARC ab initio 3D reconstruction, one class was selected with obvious huNAIP/huNLRC4 density. Two rounds of optimized 3D heterogeneous refinement yielded one class with clear huNAIP/huNLRC4 density, containing a total of 106,381 particles. Those particles were refined with C1 symmetry using non-uniform refinement, yielding a map at 3.9 Å resolution. Further particle polishing based on the map at 3.9 Å resolution and additional 3D classification improved the map quality and resolution to 3.6 Å. Local resolution was calculated using BlocRes implemented in cryoSPARC. The number of particles in each dataset and other details related to data processing are summarized in Table 1 and Extended Data Fig. 2.

### Model building and refinement

The initial templates of the NeedleTox, human NAIP and NLRC4 were derived from a homology-based model calculated by SWISS-MODEL[36].

The model was docked into the EM density map using Chimera[37] and followed by manual adjustment using COOT[38]. Note that the EM density of the CARD domain at the center disk was not sufficient to build the model. Each model was independently subjected to global refinement and minimization in real space using the module phenix.real_space_refine in PHENIX[39] against separate EM half maps with default parameters. The model was refined into a working half map, and improvement of the model was monitored using the free half map. Model geometry was further improved using Rosetta[40]. The geometry parameters of the final models were validated in COOT and using MolProbity[41] and EMRinger[42]. These refinements were performed iteratively until no further improvements were observed. The final refinement statistics are provided in Table 1. Model overfitting was evaluated through its refinement against one cryo-EM half map. Fourier shell correlation (FSC) curves were calculated between the resulting model and the working half map as well as between the resulting model and the free half and full maps for cross-validation. Figures were produced using PyMOL[43] and Chimera.

## MD simulation

Explicit solvent MD simulations were performed on apo NAIP to investigate its flexibility and dynamics in solution. The initial structure for the simulation was taken from the cryo-EM ternary complex. Residues 352–393 between the $BIR_3$ and NATCH domain were not resolved in the cryo-EM. Since the focus of the simulation was on the ID domain and its interaction with the C-terminal tail, we excluded the domain N-terminus of the NATCH domain. The first N-term residue in the simulation constructs was $Ala_{394}$. The simulations included three structural zinc ions as well as ATP.

Two simulations were performed: one for the WT NAIP (referred to as 'WT') and another with the nine C-terminal residues truncated (referred to as 'C-terminal truncate'). The truncated residues form the lasso-like structure by threading through residues 930–960 of the ID domain. The terminal residue in the C-terminal truncate simulation was Ile1394.

To enhance sampling of the lasso ring REST (replica exchange with solute tempering) MD[44] simulations were performed with the lasso residues 930–960 kept 'hot'. Eight replicas were used and the simulations were run at 300 K and 1 bar. Each replica was run for 100 ns using the Desmond simulation package[45] in Schrodinger 2021-1 (www.schrodinger.com). The systems were protonated at neutral pH and centered in a cubic box such that the minimum distance from any protein atom to the box wall was 10 Å. The box was solvated using simple point-charge[46] water molecules and counter ions were added to neutralize the system. OPLS4 force field[47] was used as the potential energy function for the protein. Default relaxation protocol in Maestro was employed before the production simulations.

Simulations were performed on Amazon Web Services cloud computing platform with each simulation employing 8 NVIDIA Tesla K80 GPU cards (Nvidia). The production run was 100 ns long, and the final trajectory from each replica contained 2,000 conformations saved at an interval of 50 ps. The number of atoms in the simulations were approximately 240,000 and a single simulation took approximately 8 days of wall time. A time step of 2 fs was used, and exchanges were attempted at an interval of 1.2 ps. In all simulations, the acceptance ratio of exchanges between the adjacent replicas was observed to be between 0.2 and 0.4. The results presented here correspond to the physical replica.

## Reporting summary

Further information on research design is available in the Nature Portfolio Reporting Summary linked to this article.

## Data availability

The cryo-EM maps of the human NLRC4 C11, C12 and Needle/huNAIP/huNLRC4-R288A have been deposited with accession codes (PDB 8FW2, EMD-29496), (PDB 8FW9, EMD-29498) and (PDB 8FVU, EMD-29493), respectively. Previously published models, used to interpret human NLRC4 and NAIP structures, were presented in this study along with their corresponding PDB ID codes (3JBL, 7RAV, 4KXF, 6B5B and 6K8J) and references. Other data are available from the corresponding author upon reasonable request. Source data are provided with this paper.

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

## Acknowledgements

We thank P.-G. Mason, J. Cargill and T. R. Curiba for large-scale expression support; S. Crampton, S. Bhattarai and C. Avila for technical contributions; and Accelagan Inc. for expression and purification support. Structural predictions were performed using an internal deployment of AlphaFold2, which was set up and deployed by the In Silico Discovery & External Innovation (ISDEI) team of Johnson & Johnson Innovative Medicine.

## Author contributions

R.E.M., X.Y., R.M., M.D.R., N.K. and S. Somani designed the experiments. R.E.M., X.Y., R.M., M.D.R., N.K. and S. Somani performed the experiments. X.Y. and R.E.M. analyzed the data, wrote the first draft of the paper and prepared data visualization. X.Y., R.E.M., S. Somani and B.F. wrote the paper. R.E.M., X.Y., S. Somani, R.A.S., Q.M., S.B., B.F. and S. Sharma edited the paper.

## Competing interests

R.E.M., X.Y., R.M., S. Somani, M.D.R., N.K., R.A.S., Q.M., S.B., B.F. and S. Sharma are employees of Johnson & Johnson Innovative Medicine.

## Additional information

**Extended data** is available for this paper at https://doi.org/10.1038/s41594-023-01143-z.

**Correspondence and requests for materials** should be addressed to Xiaodi Yu.

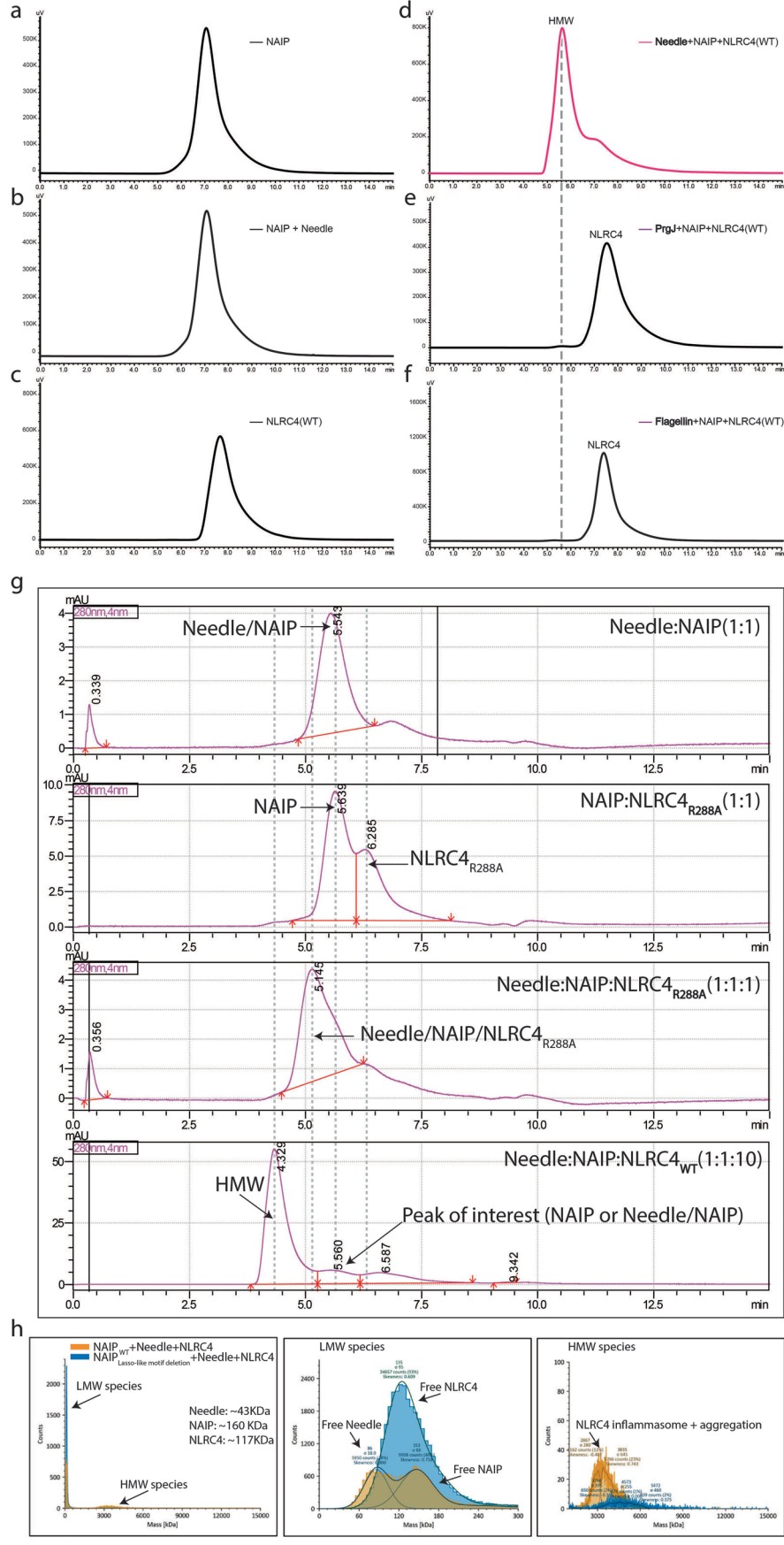

**Extended Data Fig. 1 | See next page for caption.**

**Extended Data Fig. 1 | Analytical Superdex 200 gel filtration analysis of purified proteins. a,b,c**. Chromatogram of NAIP, NAIP+Needle, and NLRC4$_{WT}$, respectively. **d,e,f**. Chromatogram of protein mixtures with various triggers. HMW species were detected in the mixture of Needle, NAIP, and NLRC4$_{WT}$. The dashed lines indicate the relative elution volume of the HMW species.
**g**. Analytica size exclusion profiles of Needle:NAIP (1:1), NAIP:huNLRC4-R288A

(1:1), Needle:NAIP:huNLRC4-R288A (1:1:1), and Needle:NAIP:huNLRC4$_{WT}$ (1:1:10) mixtures, respectively. Dashed lines indicated the relative elution time of HMW, Needle/NAIP, NAIP, or huNLRC4, respectively. **h**. Mass photometry Refeyn data of Needle:NAIP:huNLRC4$_{WT}$ (1:1:10, in orange), and Needle:$_{NAIP1-1394}$:huNLRC4 (1:1:10, in blue) mixtures. These experiments were repeated three independent times.

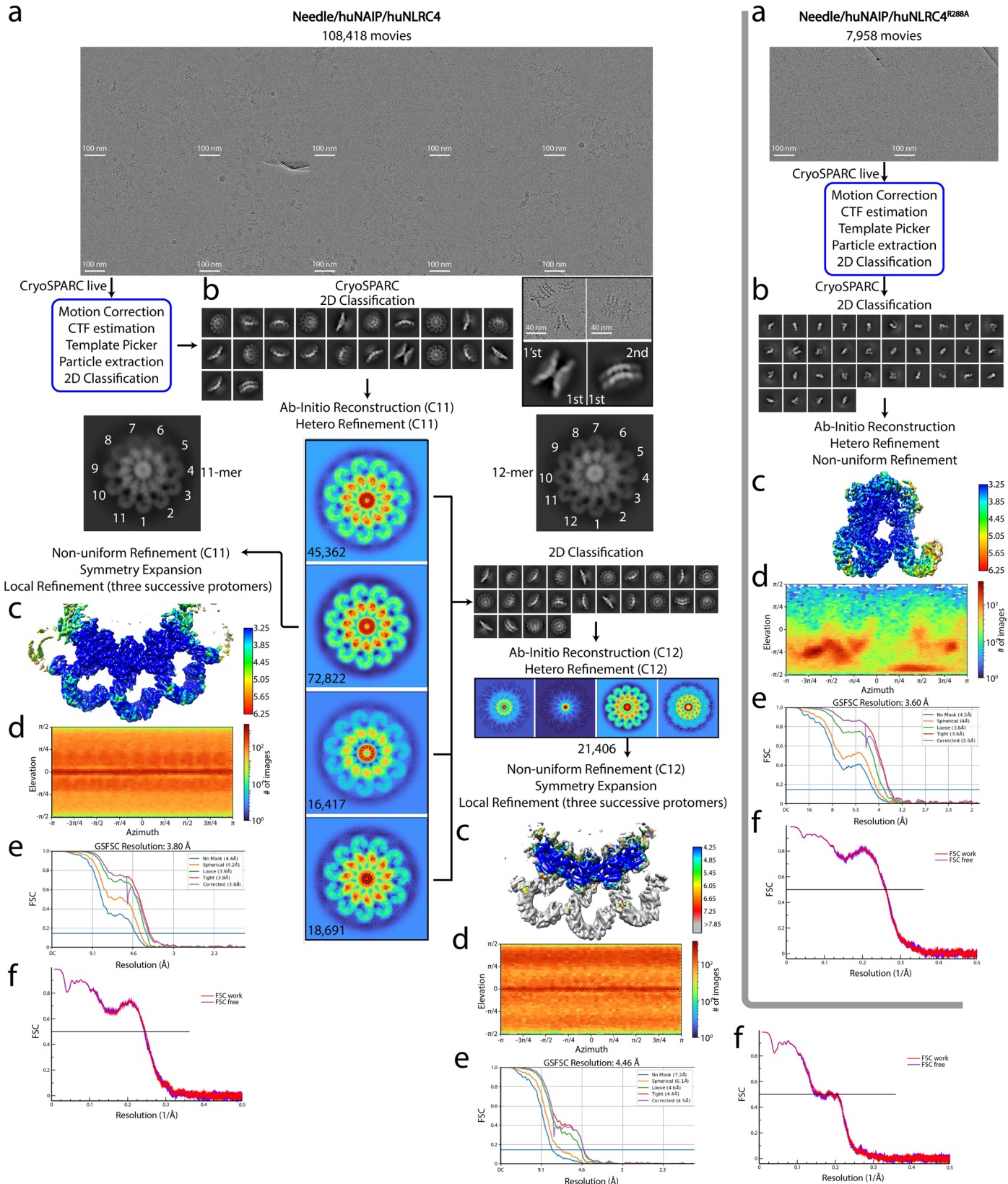

**Extended Data Fig. 2 | Cryo-EM analysis of human Needle/NAIP/NLRC 4 (Left) and Needle/NAIP/NLRC 4R288A (Right) complexes. a**. Flow chart of the cryo-EM data processing procedure. Details can be found in the Methods. **b**. Representative images and 2D classifications of inflammasome disk or disk stacks (Left), and Needle/NAIP/NLRC4R288A ternary complex (Right). **c**. Local resolution of the map and colored as indicated. **d**. Angular orientation distribution of the particles used in the final reconstruction. The particle distribution is indicated by different color shades. **e**. Fourier shell correlation (FSC) curve of the structure with FSC as a function of resolution using cryoSPARC output. f. Model validation. Comparison of the FSC curves between model and half map 1 (work), and model and half map 2 (free) are plotted in and red, respectively. These experiments were repeated three independent times.

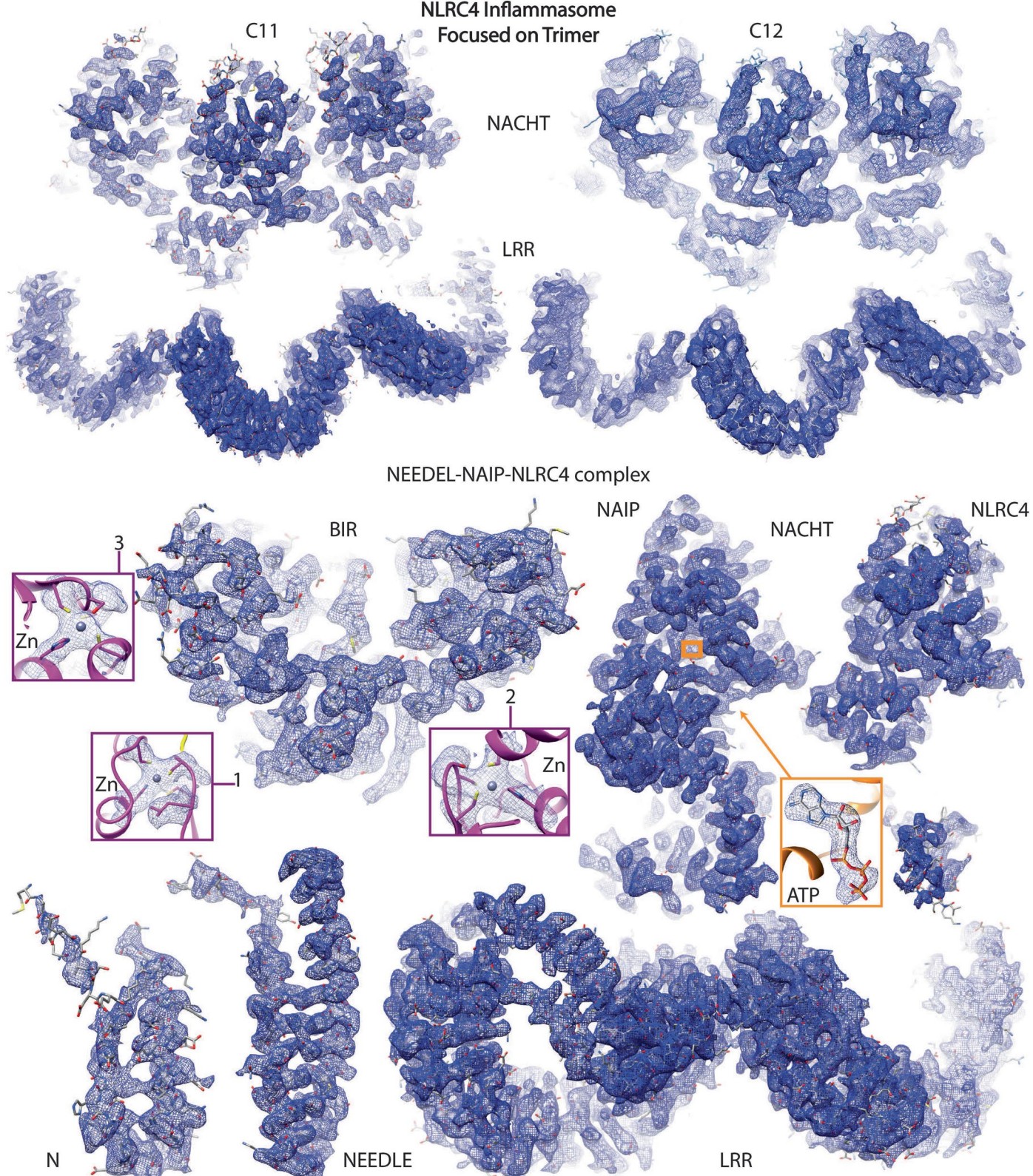

**Extended Data Fig. 3 | Cryo-EM densities of NLRC4 Triprotomer and Needle/NAIP/NLRC 4R288A complex.** Cryo-EM density is sharpened and displayed at the contour level 8σ for the highlighted regions. The atomic models with side chains are shown as sticks. The segments are labeled and colored as in Fig. 1a.

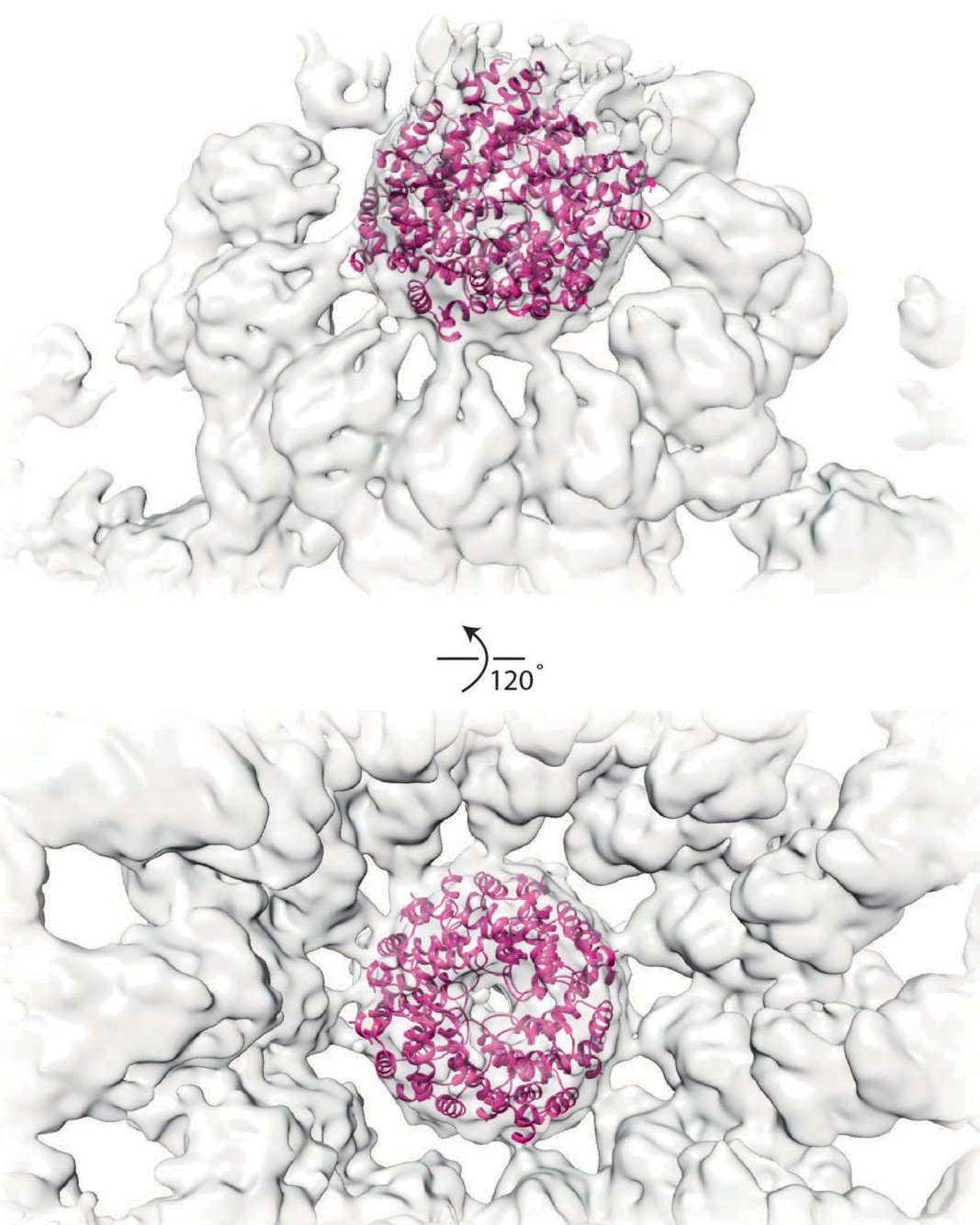

**Extended Data Fig. 4 | The CARD domain of human NLRC4 inflammasome.** The Structure of NLRC4 CARD filament (PDB ID: 6K8J) is shown as cartoon colored magenta and docked into the EM density map of human NLRC4 inflammasome with 11 protomers.

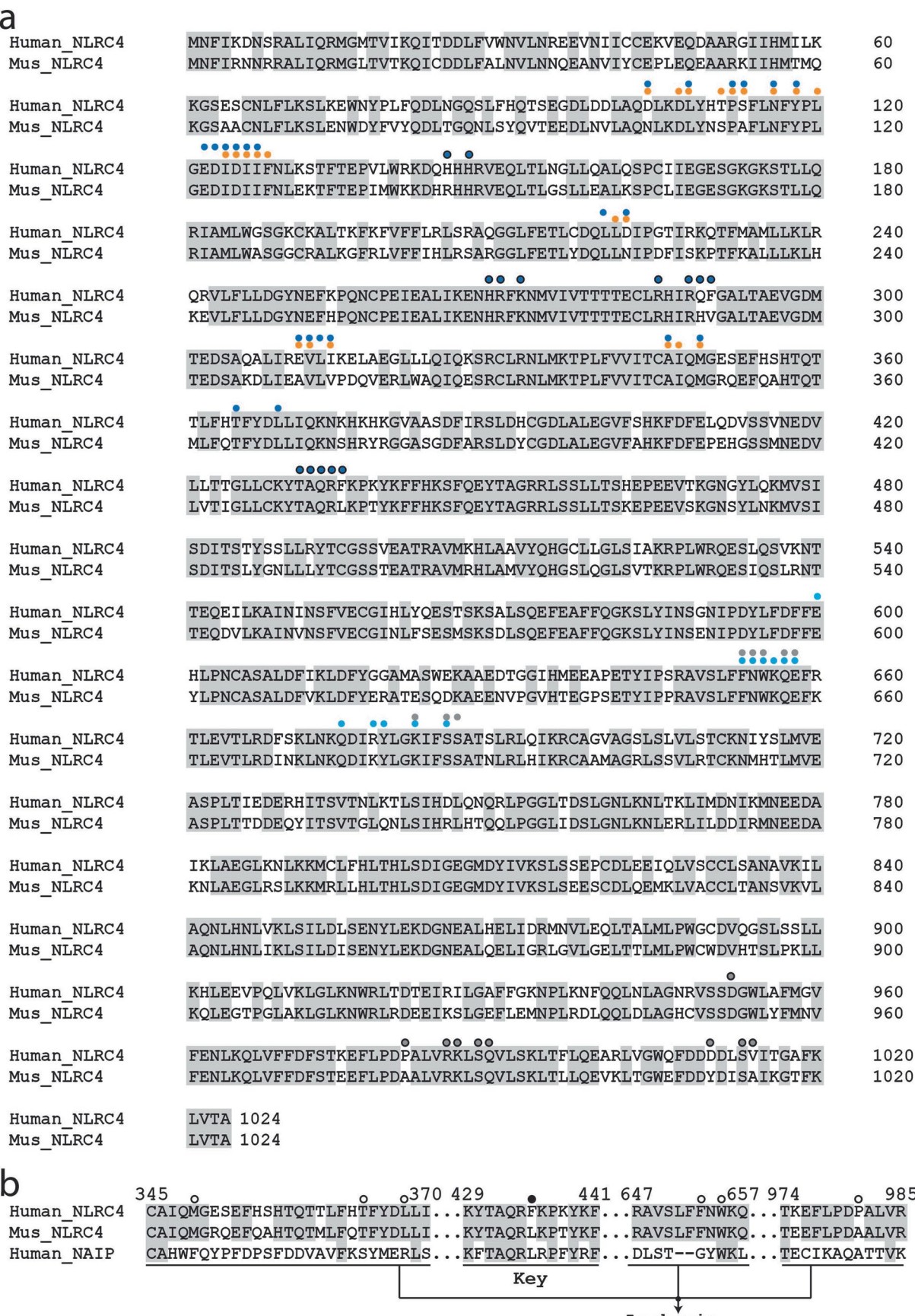

**a**

```
Human_NLRC4   MNFIKDNSRALIQRMGMTVIKQITDDLFVWNVLNREEVNIICCEKVEQDAARGIIHMILK    60
Mus_NLRC4     MNFIRNNRRALIQRMGLTVTKQICDDLFALNVLNNQEANVIYCEPLEQEAARKIIHMTMQ    60

Human_NLRC4   KGSESCNLFLKSLKEWNYPLFQDLNGQSLFHQTSEGDLDDLAQDLKDLYHTPSFLNFYPL   120
Mus_NLRC4     KGSAACNLFLKSLENWDYFVYQDLTGQNLSYQVTEEDLNVLAQNLKDLYNSPAFLNFYPL   120

Human_NLRC4   GEDIDIIFNLKSTFTEPVLWRKDQHHHRVEQLTLNGLLQALQSPCIIEGESGKGKSTLLQ   180
Mus_NLRC4     GEDIDIIFNLKEKTFTEPIMWKKDHRHHRVEQLTLGSLLEALKSPCLIEGESGKGKSTLLQ  180

Human_NLRC4   RIAMLWGSGKCKALTKFKFVFFLRLSRAQGGLFETLCDQLLDIPGTIRKQTFMAMLLKLR   240
Mus_NLRC4     RIAMLWASGGCRALKGFRLVFFIHLRSARGGLFETLYDQLLNIPDFISKPTFKALLLKLH   240

Human_NLRC4   QRVLFLLDGYNEFKPQNCPEIEALIKENHRFKNMVIVTTTTECLRHIRQFGALTAEVGDM   300
Mus_NLRC4     KEVLFLLDGYNEFHPQNCPEIEALIKENHRFKNMVIVTTTTECLRHIRHVGALTAEVGDM   300

Human_NLRC4   TEDSAQALIREVLIKELAEGLLLQIQKSRCLRNLMKTPLFVVITCAIQMGESEFHSHTQT   360
Mus_NLRC4     TEDSAKDLIEAVLVPDQVERLWAQIQESRCLRNLMKTPLFVVITCAIQMGRQEFQAHTQT   360

Human_NLRC4   TLFHTFYDLLIQKNKHKHKGVAASDFIRSLDHCGDLALEGVFSHKFDFELQDVSSVNEDV   420
Mus_NLRC4     MLFQTFYDLLIQKNSHRYRGGASGDFARSLDYCGDLALEGVFAHKFDFEPEHGSSMNEDV   420

Human_NLRC4   LLTTGLLCKYTAQRFKPKYKFFHKSFQEYTAGRRLSSLLTSHEPEEVTKGNGYLQKMVSI   480
Mus_NLRC4     LVTIGLLCKYTAQRLKPTYKFFHKSFQEYTAGRRLSSLLTSKEPEEVSKGNSYLNKMVSI   480

Human_NLRC4   SDITSTYSSLLRYTCGSSVEATRAVMKHLAAVYQHGCLLGLSIAKRPLWRQESLQSVKNT   540
Mus_NLRC4     SDITSLYGNLLLYTCGSSTEATRAVMRHLAMVYQHGSLQGLSVTKRPLWRQESIQSLRNT   540

Human_NLRC4   TEQEILKAININSFVECGIHLYQESTSKSALSQEFEAFFQGKSLYINSGNIPDYLFDFFE   600
Mus_NLRC4     TEQDVLKAINVNSFVECGINLFSESMSKSDLSQEFEAFFQGKSLYINSENIPDYLFDFFE   600

Human_NLRC4   HLPNCASALDFIKLDFYGGAMASWEKAAEDTGGIHMEEAPETYIPSRAVSLFFNWKQEFR   660
Mus_NLRC4     YLPNCASALDFVKLDFYERATESQDKAEENVPGVHTEGPSETYIPPRAVSLFFNWKQEFK   660

Human_NLRC4   TLEVTLRDFSKLNKQDIRYLGKIFSSATSLRLQIKRCAGVAGSLSLVLSTCKNIYSLMVE   720
Mus_NLRC4     TLEVTLRDINKLNKQDIKYLGKIFSSATNLRLHIKRCAAMAGRLSSVLRTCKNMHTLMVE   720

Human_NLRC4   ASPLTIEDERHITSVTNLKTLSIHDLQNQRLPGGLTDSLGNLKNLTKLIMDNIKMNEEDA   780
Mus_NLRC4     ASPLTTDDEQYITSVTGLQNLSIHRLHTQQLPGGLIDSLGNLKNLERLILDDIRMNEEDA   780

Human_NLRC4   IKLAEGLKNLKKMCLFHLTHLSDIGEGMDYIVKSLSSEPCDLEEIQLVSCCLSANAVKIL   840
Mus_NLRC4     KNLAEGLRSLKKMRLLHLTHLSDIGEGMDYIVKSLSEESCDLQEMKLVACCLTANSVKVL   840

Human_NLRC4   AQNLHNLVKLSILDLSENYLEKDGNEALHELIDRMNVLEQLTALMLPWGCDVQGSLSSLL   900
Mus_NLRC4     AQNLHNLIKLSILDISENYLEKDGNEALQELIGRLGVLGELTTLMLPWCWDVHTSLPKLL   900

Human_NLRC4   KHLEEVPQLVKLGLKNWRLTDTEIRILGAFFGKNPLKNFQQLNLAGNRVSSDGWLAFMGV   960
Mus_NLRC4     KQLEGTPGLAKLGLKNWRLRDEEIKSLGEFLEMNPLRDLQQLDLAGHCVSSDGWLYFMNV   960

Human_NLRC4   FENLKQLVFFDFSTKEFLPDPALVRKLSQVLSKLTFLQEARLVGWQFDDDDLSVITGAFK  1020
Mus_NLRC4     FENLKQLVFFDFSTEEFLPDAALVRKLSQVLSKLTLLQEVKLTGWEFDDYDISAIKGTFK  1020

Human_NLRC4   LVTA 1024
Mus_NLRC4     LVTA 1024
```

**b**

```
                345       o        o  370 429       o   441 647     o  o657 974    o    985
Human_NLRC4   CAIQMGESEFHSHTQTTLFHTFYDLLI...KYTAQRFKPKYKF...RAVSLFFNWKQ...TKEFLPDPALVR
Mus_NLRC4     CAIQMGRQEFQAHTQTMLFQTFYDLLI...KYTAQRLKPTYKF...RAVSLFFNWKQ...TEEFLPDAALVR
Human_NAIP    CAHWFQYPFDPSFDDVAVFKSYMERLS...KFTAQRLRPFYRF...DLST--GYWKL...TECIKAQATTVK
```

Key

Lock pin

**Extended Data Fig. 5 | See next page for caption.**

**Extended Data Fig. 5 | Sequence alignment of NLRC4. a.** Conserved residues are highlighted and filled in gray. Residues involved in binding are highlighted using orange and cyan circles for the NACHT-NACHT and LRR-LRR interactions with the NAIP, blue and gray circles with black outlines for the interactions with the front NLRC4, blue and gray circles without black outlines for the interactions with the back NLRC4, respectively. **b.** Sequence alignments of NLRC4 and NAIP at the 'LockKey' regions. Conserved residues are highlighted and filled in gray. Critical residues were highlighted using black and white circles with black outlines, respectively.

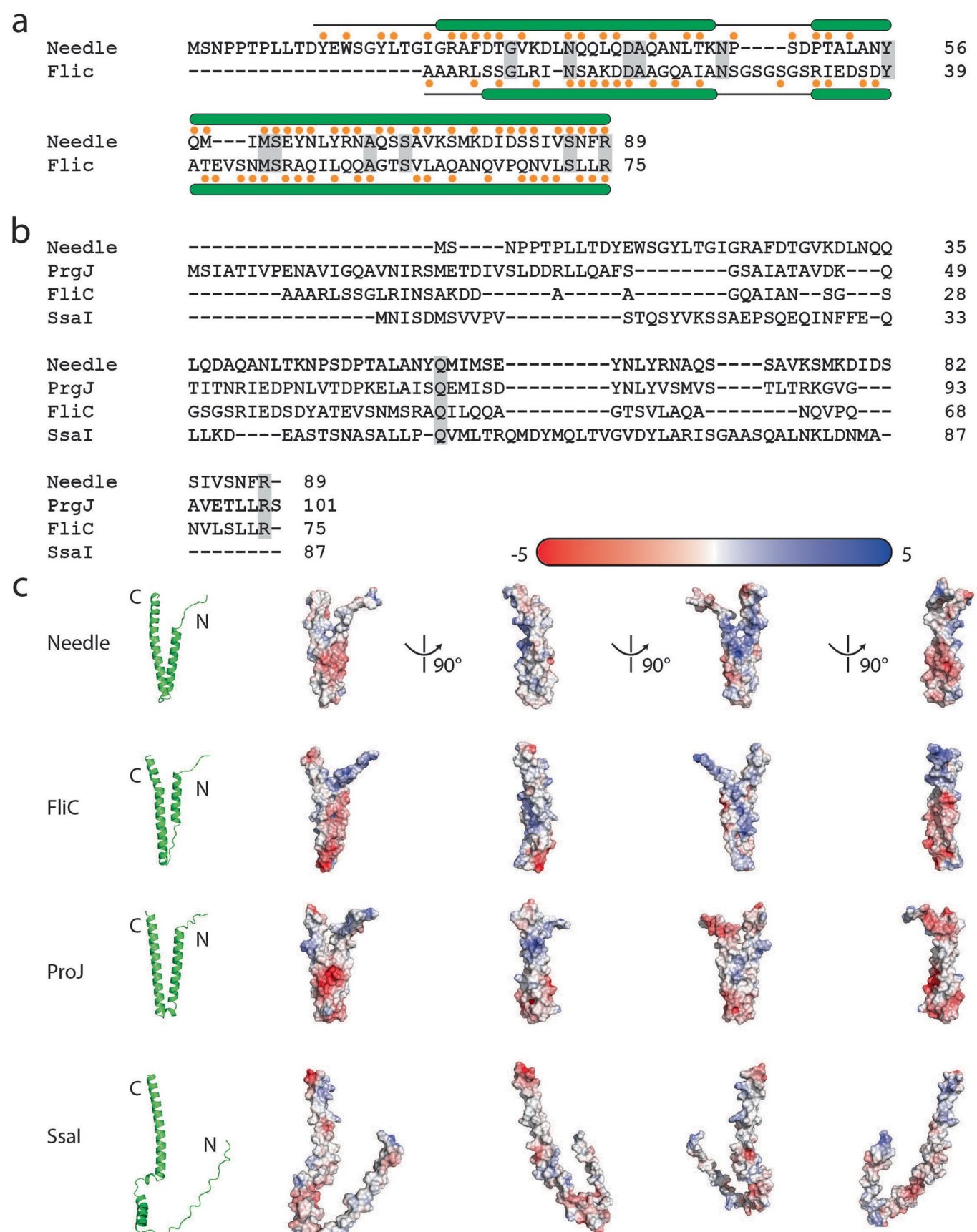

**Extended Data Fig. 6 | Sequence alignment of Needle, Flic, PrgJ, and SsaI.**
**a**. Sequence alignment of Needle and FliC. The secondary structural features are presented according to the Needle/huNAIP/huNLRC4R288A structures in this study and PDB ID:5YUD. **b**. Sequence alignment of Needle, PrgJ, FliC, and SsaI.

Conserved residues are highlighted and filled in gray. **c**. Electrostatic surface charges of Needle, PrgJ, FliC, and SsaI. The structures of Needle and FliC are from this study and PDB ID: 5YUD. The structures of PrgJ and SsaI are AlphaFold2 predicted models.

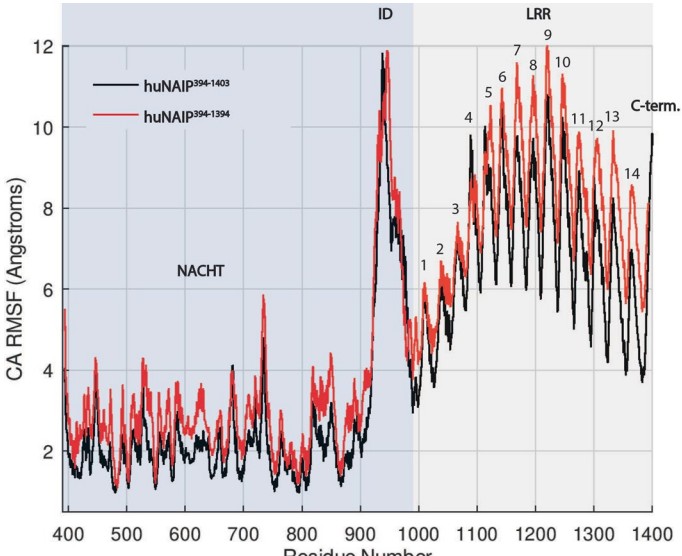

**Extended Data Fig. 7 | Molecular Dynamic simulations.** RMSF (Root Mean Squared Fluctuation) of the CA atoms for the huNAIP$_{394-1403}$ (black) and huNAIP394-1394 (red) simulations computed after aligning the trajectories on NACHT domain from the Cryo-EM structure. The fluctuation pattens are following the domain boundaries. Since the alignment was performed on the NACHT domain, for both simulations, the fluctuations in the NACHT domain are lower than in the other domains. However, the fluctuations of NACHT and LRR domains in huNAIP394-1394 simulation are higher than those in huNAIP simulation. Overall, the simulations suggest that perturbing the lasso-like structure in the C-term Truncate destabilizes the whole protein. We further inspected the flexibility of the lasso-like ring (residues 930-960) in the ID domain. Although the RMSF plot shows similar fluctuations in the ID domain, aligning the trajectories on the CA atoms of the lasso-like ring shows that the ring is substantially more flexible in huNAIP$_{1-1394}$.

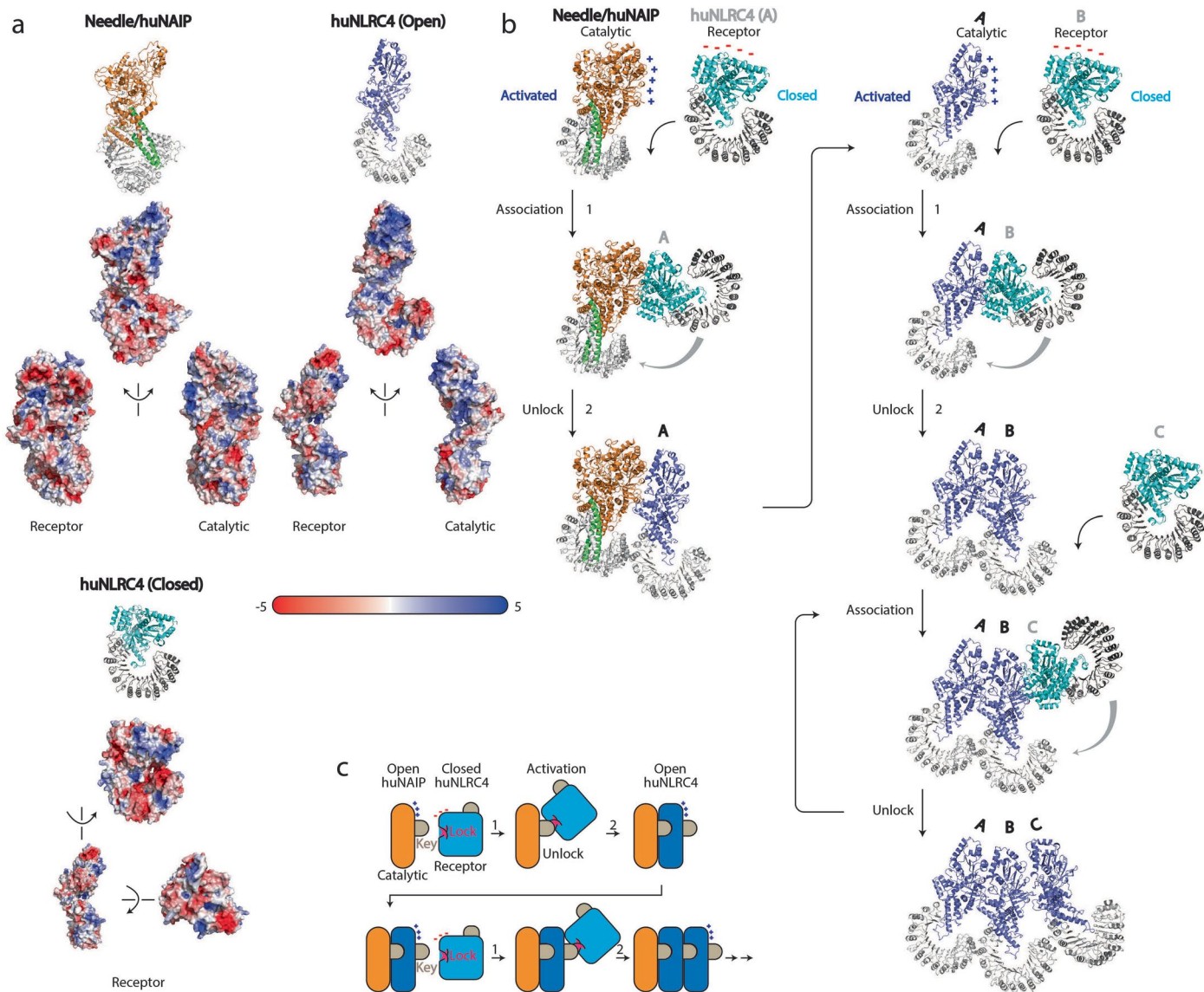

**Extended Data Fig. 8 | Closed and Open huNLRC4. a.** Different views of the electrostatic surface potentials of Needle/huNAIP, open and closed huNLRC4 calculated with the APBS. **b.** Proposed steps of huNLRC4 activation. The closed huNLRC4 was derived from the closed state of muNLRC4. NACHT is in blue or cyan for the Activated or Closed huNLRC4, respectively. LRR is in light or dark gray for the Activated and Closed huNLRC4, respectively. 1. Association step, the closed huNLRC4 associates with the activated huNLRC4 via the charge-charge surface interaction on the NACHT domains. 2. Unlock step (also in c), the 'Key' from the activated huNLRC4 inserts into the 'Lock' on the closed huNLRC4, triggering the conformational changes. The NACHT unlocking allows large movement of LRR, establishing a second binding surface between LRRs. The arrows highlight the movements during activation. **c.** Schematic representation of proposed 'Key and Lock' unlocking mechanism. 1. Association step and 2. Unlock step.

# Reporting Summary

## Statistics

For all statistical analyses, confirm that the following items are present in the figure legend, table legend, main text, or Methods section.

| n/a | Confirmed | |
|---|---|---|
| ☐ | ☒ | The exact sample size (*n*) for each experimental group/condition, given as a discrete number and unit of measurement |
| ☐ | ☒ | A statement on whether measurements were taken from distinct samples or whether the same sample was measured repeatedly |
| ☒ | ☐ | The statistical test(s) used AND whether they are one- or two-sided<br>*Only common tests should be described solely by name; describe more complex techniques in the Methods section.* |
| ☒ | ☐ | A description of all covariates tested |
| ☒ | ☐ | A description of any assumptions or corrections, such as tests of normality and adjustment for multiple comparisons |
| ☒ | ☐ | A full description of the statistical parameters including central tendency (e.g. means) or other basic estimates (e.g. regression coefficient) AND variation (e.g. standard deviation) or associated estimates of uncertainty (e.g. confidence intervals) |
| ☒ | ☐ | For null hypothesis testing, the test statistic (e.g. *F*, *t*, *r*) with confidence intervals, effect sizes, degrees of freedom and *P* value noted<br>*Give P values as exact values whenever suitable.* |
| ☒ | ☐ | For Bayesian analysis, information on the choice of priors and Markov chain Monte Carlo settings |
| ☒ | ☐ | For hierarchical and complex designs, identification of the appropriate level for tests and full reporting of outcomes |
| ☒ | ☐ | Estimates of effect sizes (e.g. Cohen's *d*, Pearson's *r*), indicating how they were calculated |

*Our web collection on statistics for biologists contains articles on many of the points above.*

## Software and code

Policy information about availability of computer code

| Data collection | EPU 3.3.0 |
|---|---|
| Data analysis | CryoSparc 4.0, Chimera 1.18, Phenix 1.18.2-3874, Coot 0.9.6, Pymol 2.5.0, Rosetta 3.13, Schrodinger 2021-1, MolProbity, EMRinger, Swiss-model 2019 |

For manuscripts utilizing custom algorithms or software that are central to the research but not yet described in published literature, software must be made available to editors and reviewers. We strongly encourage code deposition in a community repository (e.g. GitHub). See the Nature Portfolio guidelines for submitting code & software for further information.

## Data

Policy information about availability of data

All manuscripts must include a data availability statement. This statement should provide the following information, where applicable:
- Accession codes, unique identifiers, or web links for publicly available datasets
- A description of any restrictions on data availability
- For clinical datasets or third party data, please ensure that the statement adheres to our policy

The cryo-EM maps of the human NLRC4 C11, C12 and Needle/huNAIP/huNLRC4R288A have been deposited with accession codes (PDB-8FW2, EMD-29496), (PDB-8FW9, EMD-29498) and (PDB-8FVU, EMD-29493), respectively. Previously published models, used to interpret human NLRC4 and NAIP structures, were

presented in this study along with their corresponding PDB ID codes and references. Other data are available from the corresponding author upon reasonable request.

## Human research participants

Policy information about [studies involving human research participants and Sex and Gender in Research.](studies involving human research participants and Sex and Gender in Research.)

| Reporting on sex and gender | n/a |
|---|---|
| Population characteristics | n/a |
| Recruitment | n/a |
| Ethics oversight | n/a |

Note that full information on the approval of the study protocol must also be provided in the manuscript.

# Field-specific reporting

Please select the one below that is the best fit for your research. If you are not sure, read the appropriate sections before making your selection.

☒ Life sciences          ☐ Behavioural & social sciences          ☐ Ecological, evolutionary & environmental sciences

For a reference copy of the document with all sections, see [nature.com/documents/nr-reporting-summary-flat.pdf](nature.com/documents/nr-reporting-summary-flat.pdf)

# Life sciences study design

All studies must disclose on these points even when the disclosure is negative.

| | |
|---|---|
| Sample size | A significant number of cryo-EM datasets (n=6) were collected to achieve the necessary resolutions for modeling. In our case, resolution limit was attained for the CARD domain because of flexibility issues. For all biochemical experiments, sample size was determined as described in the Methods section, whenever applicable. |
| Data exclusions | Particles were automatically picked. Images with contaminations were deselected. During the process of classification and refinement, denatured or damaged protein particles were removed. For the biochemical experiments, no data were excluded from our analyses. |
| Replication | Inflammasome assembly, purification, and analysis of cryo-EM samples were repeated at least three times. |
| Randomization | This study is not involved in Animal or clinical trials. No randomization was performed. All particles in the raw images were selected and analyzed to ensure a unbiased initial particle population. After removal of damaged particles, the dataset is randomly separated into two subsets that are analyzed separately to determine the spatial resolution. |
| Blinding | This study is not involved in Animal or clinical trials. Blinding was not a requisite for cryo-EM structural studies as the assay results are not subject to bias. During data processing, particles were grouped into two separate datasets to asses the resolution limit and preformed automatically by the program and does not imply any human supervision or selection during final refinement. |

# Reporting for specific materials, systems and methods

We require information from authors about some types of materials, experimental systems and methods used in many studies. Here, indicate whether each material, system or method listed is relevant to your study. If you are not sure if a list item applies to your research, read the appropriate section before selecting a response.

## Materials & experimental systems

| n/a | Involved in the study |
|---|---|
| ☒ | ☐ Antibodies |
| ☐ | ☒ Eukaryotic cell lines |
| ☒ | ☐ Palaeontology and archaeology |
| ☒ | ☐ Animals and other organisms |
| ☒ | ☐ Clinical data |
| ☒ | ☐ Dual use research of concern |

## Methods

| n/a | Involved in the study |
|---|---|
| ☒ | ☐ ChIP-seq |
| ☒ | ☐ Flow cytometry |
| ☒ | ☐ MRI-based neuroimaging |

# Eukaryotic cell lines

Policy information about cell lines and Sex and Gender in Research

| | |
|---|---|
| Cell line source(s) | Sf9 purchased from Expression Systems LLC |
| Authentication | Authentication is carried out by the manufacture, and the successful protein expression from these strains validates their authenticity. |
| Mycoplasma contamination | No mycoplasma contamination was observed. |
| Commonly misidentified lines<br>(See ICLAC register) | No commonly misidentified cell lines were used in the study. |

