## [Peer Review File · Nature Structural & Molecular Biology]

Peer Review Information

Manuscript Title: Structural basis of the human NAIP/NLRC4 inflammasome assembly and pathogen sensing

Corresponding author name(s): Xiaodi Yu

Reviewer Comments & Decisions:

Decision Letter, initial version:

Message: 25th May 2023

Dear Dr. Yu,

Thank you again for submitting your manuscript "Structural basis of the human NAIP/NLRC4 inflammasome assembly and pathogen sensing". We now have comments (below) from the 2 reviewers who evaluated your paper. In light of those reports, we remain interested in your study and would like to see your response to the comments of the referees, in the form of a revised manuscript.

You will see that while reviewers appreciate the results, they raise several concerns which will need to be addressed in a revision. Specifically, in line with reviewers' #1 comments, we would expect the proposed mechanisms to be validated by mutagenesis and functional assays. We agree that adding cell based data would strengthen the manuscript. We agree with reviewer #2 that further investigation and/or discussion on the release mechanism would be interesting, if feasible.

Please be sure to address/respond to all concerns of the referees in full in a point-by-point response and highlight all changes in the revised manuscript text file. If you have comments that are intended for editors only, please include those in a separate cover letter.

We expect to see your revised manuscript within 6 weeks. If you cannot send it within this time, please contact us to discuss an extension; we would still consider your revision, provided that no similar work has been accepted for publication at NSMB or published

elsewhere.

Reporting Summary:

Please note that all key data shown in the main figures as cropped gels or blots should be presented in uncropped form, with molecular weight markers. These data can be aggregated into a single supplementary figure item. While these data can be displayed in a relatively informal style, they must refer back to the relevant figures. These data should be submitted with the final revision, as source data, prior to acceptance, but you may want to start putting it together at this point.

SOURCE DATA: we urge authors to provide, in tabular form, the data underlying the graphical representations used in figures. This is to further increase transparency in data reporting, as detailed in this editorial (<http://www.nature.com/nsmb/journal/v22/n10/full/nsmb.3110.html>). Spreadsheets can be submitted in excel format. Only one (1) file per figure is permitted; thus, for multi-paneled figures, the source data for each panel should be clearly labeled in the Excel file; alternately the data can be provided as multiple, clearly labeled sheets in an Excel file. When submitting files, the title field should indicate which figure the source data pertains to. We encourage our authors to provide source data at the revision stage, so that they

are part of the peer-review process.

Data availability: this journal strongly supports public availability of data. All data used in accepted papers should be available via a public data repository, or alternatively, as Supplementary Information. If data can only be shared on request, please explain why in your Data Availability Statement, and also in the correspondence with your editor. Please note that for some data types, deposition in a public repository is mandatory - more information on our data deposition policies and available repositories can be found below: <https://www.nature.com/nature-research/editorial-policies/reporting-standards#availability-of-data>

[redacted]

Sincerely,

Katarzyna Ciazynska
(she/her)

Associate Editor
Nature Structural & Molecular Biology
<https://orcid.org/0000-0002-9899-2428>

Referee expertise:

Referee #1: signalling, infection

Referee #2: cryo-EM, signalling

Reviewers' Comments:

Reviewer #1:

Remarks to the Author:

NAIP–NLRC4 inflammasome is one of the best characterized NLR-mediated inflammasome complexes functioning in innate sensing of bacterial pathogens. In the NAIP–NLRC4 inflammasome, the NAIP proteins are the bona fide receptors that directly bind bacterial ligands such as flagellin, T3SS rod, or needle proteins, whereas NLRC4 acts as an adaptor that is recruited by ligand-activated NAIPs and organizes the large inflammasome complexes to trigger pyroptosis and inflammatory cytokines maturation. Recently, elegant structural studies by single particle Cryo-EM methods uncovered the molecular mechanism for mouse NAIP–NLRC4 inflammasome assembly. One molecule of mouse NAIP2 and NAIP5 is activated by bacterial rod protein and flagellin binding, respectively, which initiates self-propagation of multiple NLRC4 molecules to assemble into oligomeric NAIP–NLRC4 inflammasome complexes. Specific recognition of bacterial ligands by different mouse NAIPs were also elucidated biochemically. Whether human NAIP–NLRC4 inflammasome shares the conserved assembly mechanism and how human NAIP recognize its bacterial ligand remains to be investigated. In this manuscript, Matico et al. report the cryo-EM structures of oligomeric human NAIP–NLRC4 inflammasome in response to bacterial T3SS needle protein as well as a ternary complex comprising the needle protein, NAIP and NLRC4 by introducing a point mutation to block NLRC4 propagation. Their experiments are well performed. The structural data are impressive and elegant, providing valuable insights into the detailed mechanisms for human NAIP–NLRC4 inflammasome assembly, i.e. ligand binding-induced activation of human NAIP and the subsequent priming of NLRC4. The manuscript is a strong candidate for Nature structural and molecular biology, but I do have some comments and suggestions for the authors to address, which shall improve the quality and the significance of their study.

1) Most of their findings are based on inferences of the structural data, including the “Lock-Key” model for human NLRC4 activation, recognition of the T3SS needle protein by human NAIP, and lasso-like motif-dependent conformational changes for NAIP activation. One major criticism is that the authors did not carry out necessary mutagenesis and cell-based functional experiments to verify their structural findings. These are important to validate the structure-derived mechanisms and strengthen the conclusions drawn.

2) Human NAIP, as well as mouse NAIP1, prefer to recognize the T3SS needle protein as its cognate ligand, albeit some studies reported that human NAIP could be activated in

cells by other ligands such as bacterial flagellin or the T3SS rod protein. The authors demonstrated in the reconstitution system that only the needle protein could induce the oligomeric assembly of human NAIP–NLRC4 inflammasome, and the solved structure of the ternary complex provides direct evidences on how human NAIP recognizes the T3SS needle protein. On this critical issue (in the inflammasome field), the authors should perform in-depth analyses of the determined structure and give more specific explanations on the ligand selectivity or preference of human NAIP. This will help to understand the functionality of human NAIP in detecting bacterial ligands during infection.

3) The authors proposed three NAIP–NLRC4 inflammasome assembly models. Model 1 and model 3 are both hetero-oligomeric complexes, consistent with previous observations in mouse NAIP–NLRC4 inflammasomes in which one ligand-bound NAIP molecule is present in the inflammasome complexes and primes NLRC4 self-propagation to form a ring or helical assembly. As for model 2, are there any experimental data in the reconstitution or cellular system supporting that NLRC4 can form a homo-oligomeric inflammasome complex by releasing the starting activated NAIP? If not, I would suggest to remove this model from the discussion, to avoid misleading or confusing researchers in the field.

4) The authors propose that the needle protein dissociates from the initial recognizing NAIP upon assembly of the complete NAIP–NLRC4 inflammasome complex. This unexpected mechanism is quite interesting and also important for defining the mechanism of NAIP–NLRC4 inflammasome assembly. However, there are no direct evidences demonstrating that this is indeed the case. The authors should experimentally investigate the hypothesis by using the native mass spectrometry to measure molecular weight of the NAIP–NLRC4 inflammasome complex or using the immunogold staining of a tagged needle protein within the fully assembled inflammasome complex.

5) In figure S1, gel-filtration profiles of the interaction between human NAIP and NLRC4 in the presence of other bacterial ligands such as PrgJ or flagellin are missing.

Reviewer #3:

Remarks to the Author:

The manuscript by Matico, Yu and colleagues describes the cryoEM structures of the full-length human NAIP and human NLRC4 inflammasome complex stimulated by the *Bacillus thailandensis* T3SS needle protein. The structures reveal how the needle protein binds and changes the conformation of NAIP through a lock-key mechanism involving hydrophobic interactions, which in turn triggers conformational changes in human NLRC4 to assemble oligomers of C11 or C12 symmetry. As this is the first full-length human NAIP–NLRC4 inflammasome structure, the reported original findings should be of significant interest to the readers of NSMB.

A previous publication by the Vance group in 2017 showed the 5.2 Å resolution EM structure of murine NAIP5–NLRC4–flagellin complex, using a GFP-tagged full-length NLRC4. The CARD density was also poor but the C11 or C12 symmetry was observed similarly to the current study. A major difference is that the current study reveals needle–huNAIP–huNLRC4 (R288A) as a trimeric complex, and huNLRC4 C11/C12 oligomers without needle–huNAIP, whereas the previous study showed oligomeric murine NLRC4 in association with a flagellin–mNAIP5 complex. As shown in the 2017 study, the NLRC4

oligomeric complex may not form a closed circular disc. Did the authors observe such non-circular NLRC4 disc among the particles, and is there a possibility of identifying needle-huNAIP in such non-circular discs similar to that in the murine NAIP5-NLRC4-flagellin complex? Since the current structures are of higher resolution, this ought to be possible even if only a small fraction of the discs have needle-huNAIP. This may represent (one of) the physiological signaling complexes.

The authors propose that either the needle or the needle-huNAIP complex needs to be released from the disc to avoid clash with other NLRC4 molecules in the disc. What are the potential mechanisms to trigger such release? Are the authors hypothesizing that the NLRC4-NLRC4 interaction may be stronger than the needle-NAIP interaction with NLRC4?

One of the unique features of human NAIP protein is that it is functionally capable of recognizing a number of bacterial ligands including needle, rod and flagellin proteins. This manuscript shows data that biochemically only purified needle protein can promote the formation of the NLRC4 oligomeric complexes. The authors further rationalize that the specificity for microbial ligand recognition by human NAIP may be dependent on sequence differences among the needle, flagellin, PrgJ or SsaI ligands at their C-termini where a hydrophobic residue followed by an Arg (Fig S6) are critical for interaction. What are the hypotheses that one can come up with to explain the previously reported function of NAIP in recognizing different microbial ligands but the failure to form complexes biochemically? Post-translational modifications were mentioned in the discussion but how come the same post-translationally modified (or not modified) NAIP can be stimulated by needle but not other ligands? Are the C-terminal residues from rod and flagellin somehow obscured in the samples used in figure S6?

The authors states in the Discussion that no NAIP has been observed in either murine or human NLRC4 disc structures. This is inaccurate. The study by the Vance group shows one NAIP5-FlaA complexed with 9 NLRC4 protomers that form an open disc.

Minor issues:

The color schemes and labels in some of the figures are confusing. For example, figure 2b highlights some of the interacting residues on the "catalytic" or "receptor" sides. It is unclear what the capitalized residue labels are for, and why some residues are labeled black, purple, or red colors, with bold or regular fonts. In figure 3b, the highlighted surfaces on the left are colored purple whereas the highlighted surface on the right are colored orange. Why is this color scheme different from that in figure 2b? Again, why are residues labeled in black, purple or red? Why aren't residues F435 and L736, the supposedly "key" residues not highlighted in figures 2b and 3b?

In figure 4a, please mark the insert panel showing the lasso-like motif in the overall structure.

In figure 4c, the legend states "bars beneath the sequences" but the figure shows shaded residues, not bars.

The manuscript shows multiple figures demonstrating models of the NLRC4 inflammasome activation, in figures 4f, 5d, and S7b. It may be advisable to consolidate these into a comprehensive model. Figure 5d shows 1, 2, and 3 on the bottom without much legend to explain what they represent.

Table 1 lists three structures. But the first two (C11 and C12) do not have needle or huNAIP in the complexes so the names "Needle/huNAIP/huNLRC4" is misleading.

There are several missing references which is troubling. For example, another paper (<https://pubmed.ncbi.nlm.nih.gov/21874021/>) was published back-to-back with reference 7 on the same topic of NAIPs/NLRC4 sensing of different microbial ligands, and should be cited together with reference 7. Similarly, another paper (<https://pubmed.ncbi.nlm.nih.gov/16648852/>) was published back-to-back with reference 9 on the same topic of flagellin activation of Ipaf/NLRC4, and should be cited together with reference 9. Please check other reference errors or omissions.

Author Rebuttal to Initial comments

Reviewer #1:

Remarks to the Author:

NAIP–NLRC4 inflammasome is one of the best characterized NLR-mediated inflammasome complexes functioning in innate sensing of bacterial pathogens. In the NAIP–NLRC4 inflammasome, the NAIP proteins are the bona fide receptors that directly bind bacterial ligands such as flagellin, T3SS rod, or needle proteins, whereas NLRC4 acts as an adaptor that is recruited by ligand-activated NAIPs and organizes the large inflammasome complexes to trigger pyroptosis and inflammatory cytokines maturation. Recently, elegant structural studies by single particle Cryo-EM methods uncovered the molecular mechanism for mouse NAIP–NLRC4 inflammasome assembly. One molecule of mouse NAIP2 and NAIP5 is activated by bacterial rod protein and flagellin binding, respectively, which initiates self-propagation of multiple NLRC4 molecules to assemble into oligomeric NAIP–NLRC4 inflammasome complexes. Specific recognition of bacterial ligands by different mouse NAIPs were also elucidated biochemically. Whether human NAIP–NLRC4 inflammasome shares the conserved assembly mechanism and how human NAIP recognize its bacterial ligand remains to be investigated. In this manuscript, Matico et al. report the cryo-EM structures of oligomeric human NAIP–NLRC4 inflammasome in response to bacterial T3SS needle protein as well as a ternary complex comprising the needle protein, NAIP and NLRC4 by introducing a point mutation to block NLRC4 propagation. Their experiments are well performed. The structural data are impressive and elegant, providing valuable insights into the detailed mechanisms for human NAIP–NLRC4 inflammasome assembly, i.e. ligand binding-induced activation of human NAIP and the subsequent priming of NLRC4. The manuscript is a strong candidate for Nature structural and molecular biology, but I do have some comments and suggestions for the authors to address, which shall improve the quality and the significance of their study.

We appreciate the reviewer's recognition of our research and recommendation to further improve the quality and significance of our study.

1) Most of their findings are based on inferences of the structural data, including the “Lock-Key” model for human NLRC4 activation, recognition of the T3SS needle protein by human NAIP, and lasso-like motif-dependent conformational changes for NAIP activation. One major criticism is that the authors did not carry out necessary mutagenesis and cell-based functional experiments to verify their structural findings. These are important to validate the structure-derived mechanisms and strengthen the conclusions drawn.

Thank you for the feedback and suggestions. Based on human NLRC4 inflammasome disc, Needle/NAIP/NLRC4_{R288A} structures, and comparison with previously published mouse system structures, we observed significant conservation of the “Lock-Key” and lasso-like motifs. The critical KEY residue for NLRC4 (Leu435 in mouse, and Phe435 in human) have been extensively studied in previously published papers in the mouse system. Hu, Z. et al. in 2015 reported “mNLRC4_{L435D} mutation still displayed flagellin-induced interaction with NAIP5 but failed to form higher-order oligomeric complex.” In the human structure, we demonstrated that both NAIP and NLRC4 contain a large hydrophobic residue at this specific position. This residue is situated within a well-defined hydrophobic pocket from the adjacent NLRC4, thereby elucidating previous mutation data.

Regarding the lasso-like motif, we conducted *in vitro* inflammasome assembly and compared it to wild-type NAIP, highlighting the effects caused by its deletion. Our approach is similar to a pull-down assay but conducted on a larger scale.

We agree with the reviewer on the importance of cell-based functional experiments to validate the mechanisms derived from the structure. Our team has been actively working on establishing an internal NLRC4 cell-based system using IL-1 β signaling as the readout for these mutants. However, we have not yet achieved a successful platform. Once the platform is established, we plan to evaluate the mouse mutation data in human cells, explore trigger activation, investigate the crosstalk between NLRP3 and NLRC4 activation, conduct cryo-ET studies on ASC species in the cell, and more. We are continuing this work, but it may take a considerable amount of time to achieve success, which could result in a delay in releasing our structural analysis findings to the field. Given the time sensitivity and the broader scientific community, we would like to share our current results with the field, as they can inspire other researchers to design cellular assays that can effectively validate our proposed models.

2) Human NAIP, as well as mouse NAIP1, prefer to recognize the T3SS needle protein as its cognate ligand, albeit some studies reported that human NAIP could be activated in cells by other ligands such as bacterial flagellin or the T3SS rod protein. The authors demonstrated in the reconstitution system that only the needle protein could induce the oligomeric assembly of human NAIP–NLRC4 inflammasome, and the solved structure of the ternary complex provides direct evidences on how human NAIP recognizes the T3SS needle protein. On this critical issue (in the inflammasome field), the authors should perform in-depth analyses of the determined structure and give more specific explanations on the ligand selectivity or preference of human NAIP. This will help to understand the functionality of human NAIP in detecting bacterial ligands during infection.

In the discussion part (page 12), we have revised the explanation of how human NAIP is capable of recognizing various triggers.

Both human and mouse NAIPs share common features, including 1) a large hydrophobic interaction area with the triggers and 2) a lasso-like motif that enhances the stability of the NAIP and trigger complex while also serving as an adaptable trigger recognition platform.

Currently, we only have interaction profiles available for mNAIP5/Flagellin and huNAIP/Needle. Consequently, we are unable to fully explain why human NAIP recognizes multiple triggers solely based on the available structural information.

Needle and huNAIP can form stable binary complexes enabling structural studies. However, we cannot disregard the possibility that weaker binding triggers such as RodTox or FlaTox might also be sufficient to activate NLRC4 inflammasome assembly in human macrophages.

Additionally, broad recognition was observed in the human primary cells but not in THP-1 and U937 cell lines. In this case, we cannot exclude the influence of the huNAIP isoform that is expressed in primary cells but not in THP-1 and U937 cell lines. Moreover, the post-translational modifications (phosphorylation and potentially ubiquitination) in human cell types, may differ from the mouse system. These modifications and their interactions with other regulators could enhance the broad recognition of human NAIP towards various triggers.

3) The authors proposed three NAIP–NLRC4 inflammasome assembly models. Model 1 and model 3 are both hetero-oligomeric complexes, consistent with previous observations in mouse NAIP–NLRC4 inflammasomes in which one ligand-bound NAIP molecule is present in the inflammasome complexes and primes NLRC4 self-propagation to form a ring or helical assembly. As for model 2, are there any experimental data in the reconstitution or cellular system supporting that NLRC4 can form a homo-oligomeric inflammasome complex by releasing the starting activated NAIP? If not, I would suggest to remove this model from the discussion, to avoid misleading or confusing researchers in the field.

Thank you for the feedback and suggestions. We agree with the reviewer's point regarding the proposed Models in Figure 5d. We hope that our analysis can serve as inspiration for both us and our peers to conduct further validation in the future.

NAIP contains three BIR domains adjacent to the NACHT domain, which can act as a useful marker to identify the position of Needle/NAIP or NAIP within the fully closed NLRC4 inflammasome disc. However, despite extensive EM data processing involving focus local refinement, 3D classification, and single subtraction, we did not observe any classes exhibiting features corresponding to the NAIP-BIRs. This suggests that NAIP may not be present in the fully assembled closed disc. Interestingly, NAIP has been identified in the partially assembled open mNAIP5–NLRC4–flagellin inflammasome (Vance group in 2017). Considering these findings, we propose that Needle/NAIP might be displaced during the incorporation of the last NLRC4 molecule into the disc to achieve full closure of the assembled disc (Model 2).

Based on our structural comparison, Needle/NAIP undergoes significant conformational changes to integrate into the fully assembled closed disc. This process could result in the release of the Needle from the Needle/NAIP complex (Model 1) or the detachment of the Needle/NAIP complex from the disc (Model 2). We believe both scenarios are possible because the structure reveals extensive hydrophobic interactions between Needle and NAIP. These strong interactions may lead to the displacement of the Needle/NAIP complex by the final NLRC4 molecule during disc assembly, ultimately sealing the disc.

Additionally, a lagging shoulder (peak of interest) was observed in the high molecular weight (HMW) fraction (Fig. S1g and attached here) during the analytical Superdex 200 gel filtration analysis of Needle/NAIP/NLRC4(WT). The peak of interest aligns with NAIP (or Needle/NAIP), smaller than Needle/NAIP/NLRC4(R288A), and larger than free NLRC4. We also detected free NAIP using mass photometry Refeyn experiment (see next question for details). These findings suggest that this shoulder may represent the free NAIP, Needle/NAIP complex, or a mixture of both. However, it is not straightforward to conclude that this observed free NAIP or Needle/NAIP is the complex displaced from the fully assembled closed disc. This is because stacks of discs were also observed from the HMW peak, which may contain a higher NLRC4 to NAIP ratio in the complex. Therefore, further investigation is required to validate the proposed models. Our proposed Model 2 provides new insights into the field, but we acknowledge the need for further validation as suggested by the reviewer. We are open to removing it from this study if necessary.

4) The authors propose that the needle protein dissociates from the initial recognizing NAIP upon assembly of the complete NAIP-NLRC4 inflammasome complex. This unexpected mechanism is quite interesting and also important for defining the mechanism of NAIP-NLRC4 inflammasome assembly. However, there are no direct evidences demonstrating that this is indeed the case. The authors should experimentally investigate the hypothesis by using the native mass spectrometry to measure molecular weight of the NAIP-NLRC4 inflammasome complex or using the immunogold staining of a tagged needle protein within the fully assembled inflammasome complex.

We appreciate the reviewer's feedback and suggestion. Despite multiple rounds of condition optimization in our NLRC4 inflammasome assembly mixture, the resulting product remains heterogeneous, with a significant amount of non-productive aggregations, with partial or fully assembled, and stacking discs. We have collected a substantial amount of data to obtain high-resolution structures. We tested the high molecular weight species using a mass photometry Refeyn experiment, revealing significant sample heterogeneity (results have been attached). Native mass spectrometry may not accurately capture the molecular weight of the target complex or the molecular weight difference associated with Needle release.

Mass photometry Refeyn data reveals heterogeneity in high molecular weight (HMW) species, including NLRC4 inflammasome and aggregation. Interestingly, deletion of the lasso-like motif in NAIP leads to an abundance of free NLRC4 in low molecular weight (LMW) species. In the case of wild-type (WT) NAIP, two peaks are observed, potentially corresponding to free Needle and free NAIP, respectively. These findings support our Model 1 and Model 2, suggesting that displacement of NAIP or Needle may occur during the final step of inflammasome disc closure. NAIP with lasso-like motif deletion predominantly results in a partially open disc, with limited displacement of NAIP.

In 2015, Wu's group utilized Ni-NTA gold labeling and negative staining method to identify the colocalization of PrgJ in the partially or fully assembled NAIP2-NLRC4 inflammasome. However, due to the limitations of the negative staining method, it is challenging to confirm whether the disc is fully or partially assembled. According to our model, we hypothesize that the removal of the trigger with or without NAIP may occur during the final step of inflammasome assembly, resulting in a fully assembled closed inflammasome, while the NAIP/trigger complex may still be present in the partial disc as shown from a previous publication by the Vance group in 2017. To accurately visualize the inflammasome and determine if the NAIP/trigger complex is present in the fully assembled closed disc, a higher-resolution technique such as cryo-electron tomography (cryo-ET) may be helpful.

We are currently optimizing conditions to visualize the human NLRC4 inflammasome and ASC speck using cryo-electron tomography (cryo-ET). We hope to share our results in our upcoming study. In our current manuscript, we have introduced this sentence on page 13 "A technique such as cryo-electron tomography (cryo-ET) may be helpful to accurately visualize the inflammasome and determine if the Needle/NAIP complex is present in the fully assembled closed disc."

5) In figure S1, gel-filtration profiles of the interaction between human NAIP and NLRC4 in the presence of other bacterial ligands such as PrgJ or flagellin are missing.

The figure and legend have been updated.

Reviewer #3:

Remarks to the Author:

The manuscript by Matico, Yu and colleagues describes the cryoEM structures of the full-length human NAIP and human NLRC4 inflammasome complex stimulated by the *Bacillus thailandensis* T3SS needle protein. The structures reveal how the needle protein binds and changes the conformation of NAIP through a lock-key mechanism involving hydrophobic interactions, which in turn triggers conformational changes in human NLRC4 to assemble oligomers of C11 or C12 symmetry. As this is the first full-length human NAIP-NLRC4 inflammasome structure, the reported original findings should be of significant interest to the readers of NSMB.

We appreciate the reviewer's valuable feedback and constructive comments on our manuscript regarding the first full-length human NAIP-NLRC4 inflammasome structure. Below please see our specific answers to these comments:

A previous publication by the Vance group in 2017 showed the 5.2 Å resolution EM structure of murine NAIP5-NLRC4-flagellin complex, using a GFP-tagged full-length NLRC4. The CARD density was also poor but the C11 or C12 symmetry was observed similarly to the current study. A major difference is that the current study reveals needle-huNAIP-huNLRC4 (R288A) as a trimeric complex, and huNLRC4 C11/C12 oligomers without needle-huNAIP, whereas the previous study showed oligomeric murine NLRC4 in association with a flagellin-mNAIP5 complex. As shown in the 2017 study, the NLRC4 oligomeric complex may not form a closed circular disc. Did the authors observe such non-circular NLRC4 disc among the particles, and is there a possibility of identifying needle-huNAIP in such non-circular discs similar to that in the murine NAIP5-NLRC4-flagellin complex? Since the current structures are of higher resolution, this ought to be possible even if only a small fraction of the discs have needle-huNAIP. This may represent (one of) the physiological signaling complexes.

The Vance group used GFP fusion and point mutations (F79A and D83A) to capture the mNAIP5-NLRC4-flagellin complex during inflammasome assembly. This improved the sample homology for EM data reconstruction. In our study, we utilized the full length of huNLRC4 and found heterogeneous particles in our EM micrographs. Instead of introducing mutations, we focused on collecting a sufficient amount of data to isolate homogeneous particles. Our 2D/3D classifications did not reveal partial discs. We hypothesize that the assembly of NLRC4 discs is a rapid process, and the GFP fusion and point mutations from the Vance group's 2017 study not only reduced sample heterogeneity but also slowed down the inflammasome assembly process, enabling the isolation of partial discs. Our unpublished data using N-Term SumoStar tagged NLRC4, a smaller domain than GFP, showed a partially open disc. Removing the N-term tag, with or without the card domain, resulted in the fully assembled disc. These findings suggest that the N-terminal tagged card domain sterically hinders NLRC4 inflammasome assembly. The Vance group's study did not capture the fully assembled discs that demonstrate the interaction between the last NLRC4

and the back side of the mNAIP. This observation is particularly intriguing for our model 2, as shown in Figure 5d.

The authors propose that either the needle or the needle-huNAIP complex needs to be released from the disc to avoid clash with other NLRC4 molecules in the disc. What are the potential mechanisms to trigger such release? Are the authors hypothesizing that the NLRC4-NLRC4 interaction may be stronger than the needle-NAIP interaction with NLRC4?

Our EM data allowed us to achieve higher resolution for C11 or C12 NLRC4 discs. We performed focused local refinement to investigate additional densities corresponding to the BIR domains or the needle from the NAIP-needle complex. However, despite thorough data processing, we did not isolate any classes exhibiting the BIR or needle features. Furthermore, when comparing the structures of tri-NLRC4 from the disc and Needle-NAIP-NLRC4(R288A), we observed significant conformational changes in the BIRs and LRR of NAIP. These changes are necessary for incorporating the Needle-NAIP complex into fully assembled discs. The conformational changes may result in the release of the needle (Figure 5d, model 1), the Needle-NAIP complex (Figure 5d, model 2), or the continuous assembly of NLRC4 above the initial Needle-NAIP complex to form stacked discs (Figure 5d, model 3). Due to limited particle population, we couldn't process the stacked discs. Furthermore, the Vance group's study on GFP fusion and point mutations in the mNAIP-NLRC4 did not observe fully assembled discs. This implies that the GFP fusion and point mutations on the NLRC4-card domain might weaken NLRC4-NLRC4 interaction, impeding the replacement of the mNAIP5-NLRC4-flagellin complex and hindering complete closure of the disc or continuous assembly of stacked discs.

One of the unique features of human NAIP protein is that it is functionally capable of recognizing a number of bacterial ligands including needle, rod and flagellin proteins. This manuscript shows data that biochemically only purified needle protein can promote the formation of the NLRC4 oligomeric complexes. The authors further rationalize that the specificity for microbial ligand recognition by human NAIP may be dependent on sequence differences among the needle, flagellin, PrgJ or Ssa ligands at their C-termini where a hydrophobic residue followed by an Arg (Fig S6) are critical for interaction. What are the hypotheses that one can come up with to explain the previously reported function of NAIP in recognizing different microbial ligands but the failure to form complexes biochemically? Post-translational modifications were mentioned in the discussion but how come the same post-translationally modified (or not modified) NAIP can be stimulated by needle but not other ligands? Are the C-terminal residues from rod and flagellin somehow obscured in the samples used in figure S6?

Uncertainty exists regarding the unique recognition ability of human NAIP for multiple triggers. Initially, it was believed that hNAIP, like its murine counterpart NAIP1, only detected the T3SS needle protein. However, subsequent studies revealed that hNAIP in primary human macrophages can also sense Salmonella flagellin and inner/rod proteins from Pseudomonas aeruginosa. Kortmann et al. found that flagellin sensing in human cells requires a specific full-length isoform of hNAIP, expressed in primary cells but not in THP-1 and U937 cell lines.

Our biochemical assays indicate that Needle can form a stable complex with hNAIP for structural studies. Extensive hydrophobic interactions were observed between hNAIP and Needle, suggesting a faster association rate but weaker binding specificity that could tolerate escaping mutations or different triggers. The adjustable lasso-like loop in hNAIP provides variability at the binding site for various triggers. Furthermore, variations in trigger delivery pathways, post-translational modifications, and binding partners among different cell types may reduce NAIP's specificity in trigger recognition. Additionally, NLRP3 has been reported to interact with NLRC4 and assist in its activation in primary human macrophages. At present, we cannot fully explain why human NAIP recognizes multiple triggers based on the hNAIP-Needle complex structure. We hope our study encourages further investigation by our peers in this area.

The authors states in the Discussion that no NAIP has been observed in either murine or human NLRC4 disc structures. This is inaccurate. The study by the Vance group shows one NAIP5-FlaA complexed with 9 NLRC4 protomers that form an open disc.

We appreciate the reviewer's feedback. Our objective is to highlight that no NAIP has been observed in fully assembled closed disc structures of either murine or human NLRC4. The Vance group's study provides valuable insights into the open partial discs. Based on this, we propose a model suggesting that the Needle-NAIP complex may be replaced during the final step of NLRC4 assembly to fully close the NLRC4 inflammasome disc. We have updated the text accordingly. Page 12" NAIP was not observed in fully assembled closed disc structures of both murine and human NLRC4 inflammasomes."

Minor issues:

The color schemes and labels in some of the figures are confusing. For example, figure 2b highlights some of the interacting residues on the "catalytic" or "receptor" sides. It is unclear what the capitalized residue labels are for, and why some residues are labeled black, purple, or red colors, with bold or regular fonts. In figure 3b, the highlighted surfaces on the left are colored purple whereas the highlighted surface on the right are colored orange. Why is this color scheme different from that in figure 2b? Again, why are residues labeled in black, purple or red? Why aren't residues F435 and L736, the supposedly "key" residues not highlighted in figures 2b and 3b?

Thank you for your feedback. Initially, we intended to elaborate on the interactions of the highlighted residues (in bold) in the subsequent figures, specifically in figures 2c and 3c. However, it has become apparent that this approach has caused confusion in the interpretation.

Regarding the surface colors, we colored the surface interfaces to match their respective counterparts. For instance, in figure 2b, which illustrates the interactions between two NACHT domains of NLRC4, both interfaces were colored in slate to represent NLRC4-NACHT. In contrast, in figure 3b, the NACHT domain from NAIP was depicted in orange, and the corresponding interaction residues on NLRC4-NACHT were mapped and colored accordingly. To enhance clarity, we have updated the figure legend to aid comprehension. The sentence was introduced into the figure legends for both figure 2b, and 3b: "The

surface interfaces were colored to match their respective counterparts. The charged and “Key” residues were highlighted and colored based on their properties: blue for positive, red for negative, and yellow for hydrophobic.”

In figure 4a, please mark the insert panel showing the lasso-like motif in the overall structure.

The figure 4a has been updated.

In figure 4c, the legend states “bars beneath the sequences” but the figure shows shaded residues, not bars.

We have made updates to the legend, which now states: “The sequences passing through the ID loop in huNAIP and NAIP5 were highlighted in grey and wheat, respectively.”

The manuscript shows multiple figures demonstrating models of the NLRC4 inflammasome activation, in figures 4f, 5d, and S7b. It may be advisable to consolidate these into a comprehensive model. Figure 5d shows 1, 2, and 3 on the bottom without much legend to explain what they represent.

Thank you for providing your feedback. We have made updates to the legend descriptions to enhance the understanding of the audience.

Figure 4f. “Schematic representation of proposed huNAIP activation steps. The Apo NAIP adopts a dynamic conformation without formation of the lasso-like motif. In its open form, NAIP creates a large association interface to capture the Needle rapidly. The engagement of Needle guides the C-terminal region through the ID loop, tightening the lasso-like motif and strengthening the complex. This activated NAIP/Needle complex provides a “Key” (L736) and a large catalytic interface (Figure 3b,c) for the activation of closed huNLRC4, initiating the assembly of the human NLRC4 inflammasome.”

Figure 5d. “d. Proposed huNLRC4 inflammasome assembly model: Needle binding activates huNAIP, forming a stable Needle/huNAIP binary complex. The activated huNAIP then activates closed huNLRC4, initiating huNLRC4 inflammasome assembly (Figure 4f). In the final step as showing in the figure, the 10th huNLRC4 incorporates into the disc, interacting with NAIP from the backside. Conformational changes in NAIP are required to fully close the assembly disc, which may result in the release of the Needle from the Needle/NAIP complex (model 1), the release of the Needle-NAIP complex from the disc (model 2), or continuous stacking of NLRC4 on top of the Needle/NAIP complex to form a stacking disc (model 3).”

Figure S7b illustrates the models of NLRC4 inflammasome activation and assembly based on structures from previous and current studies.

Table 1 lists three structures. But the first two (C11 and C12) do not have needle or huNAIP in the complexes so the names “Needle/huNAIP/huNLRPC4” is misleading.

The first two titles have been updated to huNLRC4 C11 or C12 complex, respectively.

There are several missing references which is troubling. For example, another paper (<https://pubmed.ncbi.nlm.nih.gov/21874021/>) was published back-to-back with reference 7 on the same topic of NAIPs/NLRC4 sensing of different microbial ligands, and should be cited together with reference 7. Similarly, another paper (<https://pubmed.ncbi.nlm.nih.gov/16648852/>) was published back-to-back with reference 9 on the same topic of flagellin activation of Ipaf/NLRC4, and should be cited together with reference 9. Please check other reference errors or omissions.

Thank you for your feedback. We have incorporated the appropriate references into the revised version.

Decision Letter, first revision:

Message: Our ref: NSMB-A47378A

2nd Aug 2023

Dear Dr. Yu,

Thank you for submitting your revised manuscript "Structural basis of the human NAIP/NLRC4 inflammasome assembly and pathogen sensing" (NSMB-A47378A). It has now been seen by the original referees and their comments are below. The reviewers find that the paper has improved in revision, and therefore we'll be happy in principle to publish it in Nature Structural & Molecular Biology, pending minor revisions to satisfy the referees' final requests and to comply with our editorial and formatting guidelines.

Thank you again for your interest in Nature Structural & Molecular Biology. Please do not hesitate to contact me if you have any questions.

Sincerely,

Katarzyna Ciazynska
(she/her)
Associate Editor
Nature Structural & Molecular Biology
<https://orcid.org/0000-0002-9899-2428>

Reviewer #1 (Remarks to the Author):

I am happy with the revision and would like to recommend the publication of this elegant

structure story.

Reviewer #3 (Remarks to the Author):

The authors have sufficiently addressed the reviewer's concerns.

Author Rebuttal, first revision:

Reviewer #1 (Remarks to the Author):

I am happy with the revision and would like to recommend the publication of this elegant structure story.

We greatly thank and appreciate the reviewer's recognition of our work, and for comments to further improve the quality and significance of our study.

Reviewer #3 (Remarks to the Author):

The authors have sufficiently addressed the reviewer's concerns.

We appreciate the reviewer's valuable feedback and constructive comments on our manuscript regarding the first full-length human NAIP-NLRC4 inflammasome structure.

Final Decision Letter:

Message 28th Sep 2023

:

Dear Dr. Yu,

We are now happy to accept your revised paper "Structural basis of the human NAIP/NLRC4 inflammasome assembly and pathogen sensing" for publication as an Article in Nature Structural & Molecular Biology.

Your paper will be published online soon after we receive proof corrections and will appear in print in the next available issue. You can find out your date of online publication by contacting the production team shortly after sending your proof corrections. Content is published online weekly on Mondays and Thursdays, and the embargo is set at 16:00 London time (GMT)/11:00 am US Eastern time (EST) on the day of publication. Now is the time to inform your Public Relations or Press Office about your paper, as they might be interested in promoting its publication. This will allow them time to prepare an accurate and satisfactory press release. Include your manuscript tracking number (NSMB-A47378B) and our journal name, which they will need when they contact our press office.

About one week before your paper is published online, we shall be distributing a press release to news organizations worldwide, which may very well include details of your work. We are happy for your institution or funding agency to prepare its own press release, but it must mention the embargo date and Nature Structural & Molecular Biology. If you or your Press Office have any enquiries in the meantime, please contact press@nature.com.

If you have not already done so, we strongly recommend that you upload the step-by-step protocols used in this manuscript to the Protocol Exchange. Protocol Exchange is an open online resource that allows researchers to share their detailed experimental know-how. All

uploaded protocols are made freely available, assigned DOIs for ease of citation and fully searchable through nature.com. Protocols can be linked to any publications in which they are used and will be linked to from your article. You can also establish a dedicated page to collect all your lab Protocols. By uploading your Protocols to Protocol Exchange, you are enabling researchers to more readily reproduce or adapt the methodology you use, as well as increasing the visibility of your protocols and papers. Upload your Protocols at www.nature.com/protocolexchange/. Further information can be found at www.nature.com/protocolexchange/about.

Please note that *Nature Structural & Molecular Biology* is a Transformative Journal (TJ). Authors may publish their research with us through the traditional subscription access route or make their paper immediately open access through payment of an article-processing charge (APC). Authors will not be required to make a final decision about access to their article until it has been accepted. [Find out more about Transformative Journals](https://www.springernature.com/gp/open-research/transformative-journals)

Sincerely,

Katarzyna Ciazynska
(she/her)

Associate Editor
Nature Structural & Molecular Biology
<https://orcid.org/0000-0002-9899-2428>
